# Prediction and Global Sensitivity Analysis of Long-Term Deflections in Reinforced Concrete Flexural Structures Using Surrogate Models

**DOI:** 10.3390/ma16134671

**Published:** 2023-06-28

**Authors:** Wenjiao Dan, Xinxin Yue, Min Yu, Tongjie Li, Jian Zhang

**Affiliations:** 1College of Mechanical Engineering, Anhui Science and Technology University, Chuzhou 233100, China; litongjie2000@163.com; 2College of Architecture, Anhui Science and Technology University, Bengbu 233000, China; yum@ahstu.edu.cn; 3Ocean Institute, Northwestern Polytechnical University, Taicang 215400, China; jianzhang@u.nus.edu

**Keywords:** deflection prediction, reinforced concrete structures, global sensitivity analysis, surrogate models

## Abstract

Reinforced concrete (RC) is the result of a combination of steel reinforcing rods (which have high tensile) and concrete (which has high compressive strength). Additionally, the prediction of long-term deformations of RC flexural structures and the magnitude of the influence of the relevant material and geometric parameters are important for evaluating their serviceability and safety throughout their life cycles. Empirical methods for predicting the long-term deformation of RC structures are limited due to the difficulty of considering all the influencing factors. In this study, four popular surrogate models, i.e., polynomial chaos expansion (PCE), support vector regression (SVR), Kriging, and radial basis function (RBF), are used to predict the long-term deformation of RC structures. The surrogate models were developed and evaluated using RC simply supported beam examples, and experimental datasets were collected for comparison with common machine learning models (back propagation neural network (BP), multilayer perceptron (MLP), decision tree (DT) and linear regression (LR)). The models were tested using the statistical metrics $R^{2}$, RAAE, RMAE, $RMSE,\;VAF,\;PI,\;A_{10-index}\;and\;U_{95}$. The results show that all four proposed models can effectively predict the deformation of RC structures, with PCE and SVR having the best accuracy, followed by the Kriging model and RBF. Moreover, the prediction accuracy of the surrogate model is much lower than that of the empirical method and the machine learning model in terms of the RMSE. Furthermore, a global sensitivity analysis of the material and geometric parameters affecting structural deflection using PCE is proposed. It was found that the geometric parameters are more influential than the material parameters. Additionally, there is a coupling effect between material and geometric parameters that works together to influence the long-term deflection of RC structures.

## 1. Introduction

Reinforced concrete (RC) structures are widely used as the primary components of civil engineering structures due to their high strength and durability. Long-term deflection is a major concern for civil engineers when designing RC structural elements and assessing their long-term serviceability [1,2,3,4]. The deflection of RC structures may increase over time due to internal factors such as creep and shrinkage effects of the concrete and external factors such as continuous loading, elastic deformation associated with service loads and environmental influences [5]. During the service life of RC structures, local strains within component cross-sections can reach values several times greater than the initial elastic strain; this can cause undesired usability problems in structural elements with excessive deflections or crack widths, even in structures that meet code requirements. Excessive deflections can shorten the service life of RC structural elements [2] and have significant economic consequences [6]. Thus, accurate estimates of long-term deflection are essential for the structurally sound design of large-span, small-diameter RC beams [7].

Some empirical approaches have been proposed by researchers to address the above needs [2,8,9,10,11,12]. Design code methods based on mathematical formulae, including those from the American Concrete Institute (ACI) 318 [13] and Eurocode 2 [14], are highly regarded and widely used by structural engineers in structural design practice. For example, Gribniak et al. [3] presented detailed information regarding different design codes, namely, Eurocode 2, ACI 318, ACI 435 and the new Russian code SP 52-101, and analysed the long-term deflection of RC members using these codes. Some researchers have made improvements to empirical approaches to improve their applicability [2,15,16,17,18,19,20,21,22]. However, there are several problems with these methods. First, the simplified long-term deflection equations exclude key geometric and material factors of RC members [17], limiting their applicability to only simple RC members designed with the same form and working conditions [19]. Second, deflection deviations are overestimated to ensure safe predictions, and variations in load strength have a significant effect on accuracy. Finally, these methods are only applicable for estimating the instantaneous deflection of RC structural beams [18]. However, even with the recent inclusion of geometric parameters in design code equations, the estimation accuracy remains low because these relationships are expressed in a simple linear form [2,21]. As a result, there are some limitations in predicting the deflection of RC structures.

Machine learning, as a subfield of artificial intelligence, has been used to facilitate improvements and innovations in design-related problems in civil engineering [23]. Examples include automatic identification of concrete spalling [24], predicting the mechanical strength of RC materials [25], determining the punching and shear load capacity of RC slabs [26], predicting concrete strength [27,28,29], predicting the bonding capacity of fibre-reinforced polymer (FRP) to concrete interfaces [30], and forecasting the MR of modified base materials subject to wet–dry cycles [31]. Al-Zwainy, Zaki, Al-saadi and Ibraheem [32] were the first to investigate the application of artificial neural networks (ANNs) in predicting the mechanical properties of RC beams, and more recently, Pham, Ngo and Nguyen [33] reviewed the performance of various data-driven machine learning models, including ANNs, support vector regression (SVR) and integrated models, for predicting the long-term deflection of RC beams.

In addition, surrogate models (also known as metamodels) have received extensive attention from researchers and practitioners and are mathematical or numerical approximations of complex models generated by mapping from a small number of random inputs to the corresponding model outputs. During the last few years, many surrogate models have been developed in the field of civil engineering, such as polynomial response surfaces (PRS) [34], multi-variable adaptive regression spline (MARS) [35], radial basis functions (RBF) [36], Kriging models [37]/Gaussian processes [38], support vector regression (SVR) [7], polynomial chaos expansion (PCE) [39] and alternative ensembles [40]. However, surrogate models are currently devoted mainly to structural finite element model updating [41], structural damage identification [42] and structural reliability analysis [39]. Surprisingly, few studies have examined the feasibility of applying surrogate models to the prediction of long-term deflection in RC beams despite the availability of rich and practical data samples in the literature.

Global sensitivity analysis (GSA) is designed to explore the relationship between model inputs and outputs and is used to provide quantitative indices of the effect of different input variables on the model response of interest [43,44,45]. There are two main ways of calculating global sensitivity indicators: Monte-Carlo-based and surrogate-based methods. The former requires a large number of samples to assess reliable results [46], while the latter requires only a small number of samples to construct an accurate surrogate model to quickly derive the global sensitivity indices. Sudret [47] analytically derived the PCE-based global sensitivity indices formulae, and Cheng et al. [48] and Wu et al. [49] analytically derived global sensitivity indices formulae using SVR and RBF, respectively. In recent years, the application of global sensitivity analysis in the field of finite element model updating and damage identification for civil engineering structures has received much attention [50,51,52]. Surprisingly, global sensitivity analysis techniques to assess the long-term deflection affecting RC beams have not been developed or studied.

Motivated by the above analysis, this paper presents the use of surrogate models for long-term deflection prediction of RC flexible members, compares the accuracy of global sensitivity indices calculated using PCE, SVR and RBF through numerical examples, and proposes the use of PCE for global sensitivity analysis of factors affecting the deflection of RC flexible members, which help predict the long-term deflection of RC beams in advance. The method presented in this paper will provide civil engineers with a set of data-driven tools to assess the long-term availability and safety of structures.

In this study, a data-driven modelling approach for long-term deflection prediction of concrete structures using surrogate models is proposed. In particular, four well-known surrogate models are used to predict the long-term deflection of RC flexural structures, namely PCE, SVR, Kriging and RBF. All surrogate models offer good transparency because they can generate explicit mathematical formulations that better describe the physical relationships between inputs and outputs. Additionally, the use of a PCE-based global sensitivity analysis of the factors influencing the long-term deflection of concrete structures is presented, which may help designers and civil engineers predict the long-term deflection of RC beams in advance.

The remainder of this article is organised as follows. Section 2 presents the theoretical basis of surrogate models. A global sensitivity analysis is presented in Section 3. A finite element analysis of a RC simply supported beam and the collected long-term deflection dataset of RC members are analysed in Section 4. The concluding remarks and outlines of future works are summarised in Section 5. For clarity, the abbreviations in the text are listed in full in abbreviations.

## 2. Theoretical Bases of Surrogate Model

Mainstream surrogate models can be divided into regression fitting (e.g., PCE and SVR), interpolation fitting (e.g., Kriging and RBF), and a combination of both [39]. Regression fitting does not cross the training samples in the modelling process, and there is fitting error. This method can filter out noise and experimental errors in training samples and is suitable for analytical problems with some computational noise and errors. Interpolation fitting passes through all training samples during the fitting process without fitting errors. Moreover, this approach is suitable for analytical problems with small or zero error. Popular surrogate model methods in the engineering field include PCE, LSSVR, Kriging and RBF. The specific characteristics of various methods are analysed as follows.

### 2.1. PCE

PCE is an explicit representation of the stochastic model response as a series of normal multivariate polynomials [53]. First introduced into stochastic mechanics by Ghanem and Spanos, the theory of chiastic chaos [54] was later extended by Xiu and Karniadakis [55] to different types of statistical distributions (e.g., uniform, β and γ distributions), as shown in Table 1. Typically, two methods are used to solve for PCE coefficients: intrusive and non-intrusive. Intrusive methods require modifications to the solution scheme of the deterministic control equations of the model [56], while non-intrusive methods, such as projection [57] and regression [47,58], calculate the PC coefficients by repeating the simulation over a limited number of input and output samples.

The output of the physical model or system of interest can be expressed as a black-box function $y=F\left(x\right)$ of its associated variables, and this functional relationship can be expressed in the form of a polynomial chaos expansion as follows:(1)$$y=F\left(x\right)=\sum_{\alpha \in N^{n}}\beta_{\alpha }\psi_{\alpha }\left(x\right)$$
where $\alpha =\left(\alpha_{1},\ldots ,\alpha_{n}\right),(\alpha_{i}\geq 0)$ is an n-dimensional indicator, $\beta_{\alpha }$ is the unknown coefficient to be determined, and $\psi_{\alpha }$ denotes the tensor product of orthogonal polynomials in a single variable.
(2)$$\psi_{\alpha }\left(x\right)=\prod_{i=1}^{n}\psi_{\alpha_{i}}^{\left(i\right)}\left(x_{i}\right)$$

As shown in Table 1, different univariate orthogonal polynomial bases for polynomial chaos expansion can be chosen for different types of data distributions. For example, for the input variables of Gaussian distribution type, the Hermite polynomial basis can be chosen; for the input variables of a uniform distribution, the Legendre polynomial basis can be chosen.

In practical engineering applications, in order to save computational resources, the PCE in Equation (1) is usually truncated. This maintains its total order $\left|\alpha \right|=\sum_{i=1}^{n}\alpha_{i}$ while not exceeding a given polynomial of order p, i.e.,
(3)$$y≃F_{p}\left(x\right)=\sum_{\alpha \in A^{p,n}}\beta_{\alpha }\psi_{\alpha }\left(x\right),A^{p,n}=\left\{\alpha \in N^{n}:\left|\alpha \right|\leq p\right\}$$

Equation (3) is called the p-order full polynomial chaos expansion of the model response y. The relationship between the total number of unknown coefficients P and the maximum order p and dimension n of the input variables is as follows:(4)$$P=\left(\begin{array}{c} n+p\\ p \end{array}\right)=\frac{\left(n+p\right)!}{n!p!}$$

From Equation (4), the number of truncated PCE bases increases exponentially as the dimensionality of the analysed problem and the increasing order of PCE, which greatly increases the computational cost. A large number of experimental cases show that only a small number of bases in the truncated PCE have an impact on the output response. Therefore, let A be a nonempty finite subset of $N^{n}$, and the sparse PCE can be defined by the following equation:(5)$$F_{A}\left(x\right)=\sum_{\alpha \in A}\beta_{\alpha }\psi_{\alpha }\left(x\right)$$

In Equation (5), the set A is referred to as the truncated set. A truncated PCE is called sparse if the sparsity index *IS* satisfies the following conditions:(6)$$IS=\frac{\mathrm{card}\left(A\right)}{\mathrm{card}\left(A^{p_{max},n}\right)}\ll 1,p_{max}=\underset{\alpha \in A}{\max}\left(\left|\alpha \right|\right)$$
where $p_{max}$ corresponds to the order of the truncated PCE in Equation (5). In addition, the order of any indicator α in A and the order of the coupling effect of the variables are defined by the following equations, respectively:(7)$$p_{\alpha }=\left|\alpha \right|=\sum_{i=1}^{n}\alpha_{i},\eta_{\alpha }=\sum_{i=1}^{n}1_{\alpha_{i}\;>\;0}$$
where $1_{\alpha_{i}\;>\;0}=1$ if $\alpha_{i}>0$ and 0, otherwise.

A popular way of obtaining the PCE coefficients is by constructing the objective function $\underset{\beta \in R^{P}}{\min}{E\left(\beta \right)=\left\|\beta \right\|}_{1}+\lambda {\left\|\psi \beta -y\right\|}_{2}^{2}$ from Equation (5) and solving it using a greedy algorithm. A comprehensive comparison of the accuracy and efficiency of orthogonal matching pursuit (OMP), least angle regression (LAR) and Bregman-iterative greedy coordinate descent (BGCD) in solving sparse PCE was made by Zhang et al. [59]. In this paper, the BGCD algorithm will be used to solve the PCE coefficients. As soon as a sparse PCE model for the deformation of the RC structure is established, the global sensitivity index can be obtained directly by post-processing the PCE coefficients.

### 2.2. SVR

SVM is a new machine learning algorithm based on statistical learning theory developed by Vapnik et al. [60]. The SVM method adopts the principle of structural risk minimisation and integrates techniques such as convex quadratic programming, maximum interval hyperplane classification and Mercer kernel clustering, which can find the optimal compromise between the complexity of the model and the learning ability. The optimal compromise between model complexity and learning ability can avoid the problem of overfitting and falling into a local optimum in the learning process. SVM is essentially a quadratic programming problem with linear constraints and has a high training complexity. However, when the sample data are too large, SVM becomes very complex and time-consuming in solving quadratic programming problems. To overcome this problem, Suykens et al. improved the SVM model [61] by replacing the inequality constraint with an equation constraint to minimise the squared term of the error. This eventually transforms the problem into solving a set of linear equations and improves the computational efficiency and accuracy; this method is called the least squares support vector machine. When the least squares SVM is used for regression prediction and modelling, we call it SVR.

SVR is suitable for nonlinear model prediction and achieving solution sparsity, and has a wide range of application prospects. A regression function of SVR is defined as:(8)$$y\left(x\right)=\beta^{T}\varphi \left(x\right)+b$$
where b is the bias term, β denotes the regression coefficient vector and $\varphi \left(x\right)$ denotes the nonlinear mapping function that can map the original low-dimensional design variables to a high-dimensional space. The model construction can be converted into solving the optimisation problem, i.e.,
(9)J(β,e)β,eMin=12βTβ+12γ∑i=1mei2

The constraint is defined as:(10)$$y_{i}=\beta^{T}\varphi \left(x_{i}\right)+b+e_{i},\;\;i=1,2,\ldots ,m$$
where b and $e_{i}$ are the penalty factor and error variable, respectively.

By introducing Lagrange multipliers α, the optimisation problem with constraints is transformed as follows:(11)$$L\left(\beta ,b,e,\alpha \right)=\frac{1}{2}\beta^{T}\beta +\frac{1}{2}\gamma \sum_{i=1}^{m}e_{i}^{2}-\sum_{i=1}^{m}\alpha_{i}[\beta^{T}\varphi \left(x_{i}\right)+b+e_{i}-y_{i}]$$

Using the above to find the partial derivatives of the parameters β,b,e and α respectively, we obtain:(12)$$\frac{\partial L}{\partial \beta }=0\Rightarrow \beta =\sum_{i=1}^{m}\alpha_{i}\varphi \left(x_{i}\right)\;\frac{\partial L}{\partial b}=0\Rightarrow \sum_{i=1}^{m}\alpha_{i}=0\;\frac{\partial L}{\partial e}=0\Rightarrow \alpha_{i}=\gamma e_{i},i=1,2,\ldots ,m\;\frac{\partial L}{\partial \alpha }=0\Rightarrow \beta^{T}\varphi \left(x_{i}\right)+b+e_{i}-y_{i}=0,i=1,2,\ldots ,m$$

By solving the above equation and eliminating the parameters β and $e_{i}$, the following system of linear equations is obtained
(13)$$\left[\begin{array}{cc} 0 & I^{T}\\ I & \Lambda +\frac{1}{\gamma }I \end{array}\right]\left[\begin{array}{c} b\\ \alpha \end{array}\right]=\left[\begin{array}{c} 0\\ y \end{array}\right]$$
where
(14)$$\left\{\begin{array}{c} \begin{array}{c} y={\left[y_{1},y_{2},\ldots ,y_{m}\right]}^{T}\\ I={\left[1,1,\ldots ,1\right]}^{T} \end{array}\\ \begin{array}{c} \alpha ={\left[\alpha_{1},\alpha_{2},\ldots ,\alpha_{m}\right]}^{T}\\ \Lambda_{i,j}={\varphi \left(x_{i}\right)}^{T}\varphi \left(x_{j}\right) \end{array} \end{array},\;i,j=1,2,\ldots ,m\right.$$

Low-dimensional input variables can be mapped to a high-dimensional space with the following expression $\phi (x_{i},x_{j})$:(15)$$\phi \left(x_{i},x_{j}\right)={\varphi \left(x_{i}\right)}^{T}\varphi \left(x_{j}\right),\;i,j=1,2,\ldots ,m$$

Finally, the regression function of SVR is
(16)$$y\left(x\right)=\sum_{i=1}^{m}\beta_{i}\phi \left(x,x_{i}\right)+b$$

The commonly used kernel functions for constructing SVR are shown in Table 2. In this paper, we choose a radial basis kernel function with a hyperparameter σ to construct the SVR. The hyperparameter σ is obtained by cross-validation [61].

### 2.3. Kriging

As a semi-parametric model based on statistical prediction of stochastic processes, Kriging provides a linear unbiased, minimum variance estimate of the unknown response values in the design space by fitting a functional relationship between the sample points and the response values in the design space. Kriging was first proposed by the South African geologist Krige in 1951 when he was studying the distribution pattern of mineral reserves and was first used by Sacks [62] in the optimal design of structures. Kriging was first used in structural design optimisation by Simpson et al. [63,64].

Kriging consists of a parametric model and a non-parametric stochastic process jointly. For a set of *m* N-dimensional sample points, let the set consisting of sample points be $X={\left\{x_{1},x_{2},\ldots ,x_{m}\right\}}^{T}$ whose corresponding response is $Y={\left\{f\left(x_{1}\right),f\left(x_{2}\right),\ldots ,f\left(x_{m}\right)\right\}}^{T}$. Then the relationship between them can be expressed by the Kriging surrogate model as
(17)$$y\left(x\right)=f^{T}\left(x\right)\beta +z\left(x\right)$$

The first part of the equation is a linear regression of the data as shown in Equation (17), providing a global approximation to the fit, usually consisting of p polynomials. The second part is a random process with a non-independent but identically distributed normal distribution, providing a local approximation to the fit as follows:(18)$$f^{T}\left(x\right)\beta =\beta_{1}f_{1}\left(x\right)+\beta_{2}f_{2}\left(x\right)+\ldots +\beta_{p}f_{p}\left(x\right)$$

For a stochastic process $z\left(x\right)$ that is a Gaussian smooth stochastic process with non-zero covariance and subject to a normal distribution $N\left(0,\sigma^{2}\right)$, its covariance matrix is generally expressed as
(19)$$E\left[z\left(x_{i}\right),z\left(x_{j}\right)\right]=\sigma^{2}R\left(\theta ,x_{i},x_{j}\right)$$
where θ is the correlation function parameter and $R\left(\theta ,x_{i},x_{j}\right)$ is the spatial correlation function of any two sample points $x_{i},x_{j}$ in the sample points, which plays a dominant role in the fitting accuracy of the model and is commonly used in the form of a Gaussian correlation model.
(20)$$R\left[z\left(x_{i}\right),z\left(x_{j}\right)\right]=exp\left(-\sum_{k=1}^{n}\theta_{k}{\left|x_{i}^{k}-x_{j}^{k}\right|}^{2}\right)$$
where ${\left|x_{i}^{k}-x_{j}^{k}\right|}^{2}$ is the square of the distance between the two sample points in the *k*th dimension, *n* represents the total number of output parameters, and $\theta_{k}$ is the decay rate controlling the correlation in different dimensions. The correlation function matrix between each sample point in a sample set of *m* sample points is
(21)$$R=\left[\begin{array}{ccc} R\left(x_{1},x_{1}\right) & \cdots & R\left(x_{1},x_{m}\right)\\ \vdots & \ddots & \vdots \\ R\left(x_{m},x_{1}\right) & \cdots & R\left(x_{m},x_{m}\right) \end{array}\right]$$

In the above mathematical model, the likelihood function for the occurrence of the true response at the sample point can be obtained as
(22)$$L=\frac{1}{{\left(2\pi \sigma^{2}\right)}^{n/2}{\left|R\right|}^{1/2}}exp\left(-\frac{{\left(Y-F\beta \right)}^{T}R^{-1}\left(Y-F\beta \right)}{2\sigma^{2}}\right)$$
where ***F*** is a matrix of $f\left(x\right)$ vector values at each sample point.

According to the maximum likelihood rule, it can be found that:(23)$$\hat{\beta }={\left(F^{T}R^{-1}F\right)}^{-1}F^{T}R^{-1}Y\;{\hat{\sigma }}^{2}={\left(Y-F\beta \right)}^{T}R^{-1}\left(Y-F\beta \right)/m$$

Furthermore, the logarithmic form of the great likelihood function is presented as:(24)$$\ln\left(L\right)\approx -\frac{n}{2}\ln\left({\hat{\sigma }}^{2}\right)-\frac{1}{2}\ln\left|R\right|$$

The optimal solution is obtained using an optimisation algorithm, i.e., the decay rate $\theta_{k}$ in different dimensions can be determined. This method allows Kriging to be constructed. Using Kriging for unknown sample points $x_{0}$ predictions can be expressed as follows:(25)$$\hat{y}\left(x_{0}\right)=f^{T}\left(x_{0}\right)\hat{\beta }+r^{T}\left(x_{0}\right)R^{-1}\left(Y-F\beta \right)$$
where $r^{T}\left(x_{0}\right)$ is the vector of correlation functions between the unknown points and each sample point, i.e.:(26)$$r^{T}\left(x_{0}\right)=\left[R\left(x_{0},x_{1}\right),\ldots ,R\left(x_{0},x_{n}\right)\right]$$

### 2.4. RBF

RBF is a radially symmetric function, which is an interpolation method with the advantages of simplicity of form, adaptability and accuracy. It is confirmed that radial basis functions are the only best form of approximation for unknown functions by the equivalent definition of Micchelli’s theorem [65]. Frank [66] interpolated a large amount of scattered data using various interpolation methods and verified that interpolation methods based on radial basis functions were the most effective.

Suppose the function $y=f\left(x\right)$ is an N-dimensional real-valued function, and *m* sample points are selected using the experimental design method, and the set of these sample points is denoted as: $X={\left\{x_{1},x_{2},\ldots ,x_{m}\right\}}^{T}$; the set of response values $y={\left\{f\left(x_{1}\right),f\left(x_{2}\right),\ldots ,f\left(x_{m}\right)\right\}}^{T}$ for each sample point is obtained by the function. Then the radial basis function $\tilde{y}\left(x\right)$ is used to fit a function y of the following form.
(27)$$\tilde{y}\left(x\right)=\sum_{i=1}^{m}\lambda_{i}\phi \left(\left\|x-x_{i}\right\|\right)$$
where $\lambda_{i}$ represents the coefficient to be determined before the *i*th basis function and $\left\|x-x_{i}\right\|$ represents the Euclidean norm between the prediction point x and the sample point $x_{i}$. For an N-dimensional design space $\left\|x-x_{i}\right\|=\sqrt{{\left(x_{1}-x_{1}^{i}\right)}^{2}+{\left(x_{2}-x_{2}^{i}\right)}^{2}+\ldots +{\left(x_{n}-x_{n}^{i}\right)}^{2}}$, ϕ represents the radial basis form of the radial basis, and the form of the basis function usually used is shown in Table 3, such that $r=\left\|x-x_{i}\right\|$.

By taking the *m* sample points and response values into Equation (27), we obtain:(28)$$\left\{\begin{array}{c} \lambda_{1}\phi \left(r^{11}\right)+\lambda_{2}\phi \left(r^{12}\right)+\ldots +\lambda_{n}\phi \left(r^{1n}\right)=f\left(x_{1}\right)\\ \ldots \\ \lambda_{1}\phi \left(r^{m1}\right)+\lambda_{2}\phi \left(r^{m2}\right)+\ldots +\lambda_{n}\phi \left(r^{mn}\right)=f\left(x_{m}\right) \end{array}\right.$$

The above equation has a total of *m* equations and *n* unknowns. For ease of presentation, we write Equation (28) in matrix form as follows.
(29)Aλ=y
where ***A***$=\left[\begin{array}{ccc} \phi \left(r^{11}\right) & \cdots & \phi \left(r^{1n}\right)\\ \vdots & \ddots & \vdots \\ \phi \left(r^{m1}\right) & \cdots & \phi \left(r^{mn}\right) \end{array}\right]$, $\lambda ={\left\{\lambda_{1},\lambda_{2},\ldots ,\lambda_{m}\right\}}^{T},y={\left\{f\left(x_{1}\right),f\left(x_{2}\right),\ldots ,f\left(x_{m}\right)\right\}}^{T}$. Once a set of sample points and response values are given, the coefficient λ to be determined can be found according to the least squares method.

Equation (27) shows that RBF describes the complex implicit functional relationship between the structural response and the structural parameters through a linear combination of basis functions. As the number of parameters to be corrected increases, the number of sample points required to solve for the coefficients to be determined is linearly related to the number of parameters to be corrected. Thus, compared to PCE, Kriging and SVR, RBF has the advantage of saving computational costs when fitting unknown problems.

## 3. Global Sensitivity Analysis

### 3.1. Sobol’ Decomposition

Let us consider a mathematical model with an n×m input x consisting of m samples of n variables and an m×1 output y:(30)$$y=f\left(x\right),x\in K^{n}$$
where the input variables are defined on the *n*-dimensional unit cube $K^{n}$:(31)$$K^{n}=\{x:0\leq x_{i}\leq 1,\;i=1,\ldots ,n\}$$

The Sobol’ decomposition decomposes f(x) into summation terms with increasing dimensionality [44]:(32)$$f\left(x_{1},\ldots ,x_{n}\right)=f_{0}+\sum_{i=1}^{n}f_{i}(x_{i})+\sum_{1\leq i<j\leq n}f_{ij}(x_{i},x_{j})+\ldots +f_{1,2,\ldots ,n}(x_{1},\ldots ,x_{n})$$
where the constant $f_{0}$ is the mean value of the function, i.e.:(33)$$f_{0}=\int_{K^{n}}^{}f(x)dx,\;dx=dx_{1},\ldots ,dx_{n}$$

The sum in Equation (32) contains the number of summands equal to $\sum_{j=1}^{n}\left(\begin{array}{c} n\\ j \end{array}\right)=2^{n}-1$. Each summand $f_{i_{1},\ldots ,i_{s}}(x_{i_{1}},\ldots ,x_{i_{s}})$ has zero integration over any of its independent variables and the summand terms are orthogonal to each other [43], as follows:(34)$$\int_{0}^{1}f_{i_{1},\ldots ,i_{s}}(x_{i_{1}},\ldots ,x_{i_{s}})dx_{i_{k}}=0\;for\;1\leq k\leq s$$
(35)$$\int_{K^{n}}^{}f_{i_{1},\ldots ,i_{s}}\left(x_{i_{1}},\ldots ,x_{i_{s}}\right)f_{j_{1},\ldots ,j_{t}}\left(x_{j_{1}},\ldots ,x_{j_{t}}\right)dx=0\;for\{i_{1},\ldots ,i_{s}\neq j_{1},\ldots ,j_{t}\}$$

According to the above properties, the decomposition in Equation (32) is unique as long as $f\left(x\right)$ is integrable over $K^{n}$. Furthermore, the summands in the decomposition can be derived analytically. In fact, the univariate terms and the bivariate terms are represented as:(36)$$f_{i}\left(x_{i}\right)=\int_{K^{n-1}}^{}f(x)dx_{\sim i}-f_{0}$$
(37)$$f_{ij}\left(x_{i},x_{i}\right)=\int_{K^{n-2}}^{}f(x)dx_{\sim ij}-f_{i}\left(x_{i}\right)-f_{j}\left(x_{j}\right)-f_{0}$$
where $\int_{K^{n-1}}^{}dx_{\sim i}$ means integration over all variables except $x_{i}$, and $\int_{K^{n-2}}^{}dx_{\sim \{ij\}}$ means integration over all parameters except $x_{i}$ and $x_{j}$. Following this construction, any summand $f_{i_{1},\ldots ,i_{s}}(x_{i_{1}},\ldots ,x_{i_{s}})$ can be written as the difference between a multidimensional integral and a lower-order summation term.

Now consider the input parameter $X=\left\{X_{1},\ldots ,X_{n}\right\}$ as an independent random variable uniformly distributed in [0, 1]. The model response **Y** = f(***X***) is a random variable whose total variance *D* is expressed as:(38)$$D=Var\left[f\left(X\right)\right]=\int_{K^{n}}^{}f^{2}(X)dX-f_{0}^{2}$$

By integrating the square of Equation (32) and using Equation (35), the total variance in Equation (38) can be decomposed as follows:(39)$$D=\sum_{i=1}^{n}D_{i}+\sum_{1\leq i<j\leq n}D_{ij}+\ldots +D_{1,2,\ldots ,n}$$
where the partial variance appearing in the above expansion is as follows:(40)$$D_{i_{1},\ldots ,i_{s}}=\int_{K^{s}}^{}f_{i_{1},\ldots ,i_{s}}^{2}\left(x_{i_{1}},\ldots ,x_{i_{s}}\right)dx_{i_{1}},\ldots ,dx_{i_{s}},\;1\leq i_{1}<\ldots <i_{s}\leq n,s=1,\ldots ,n$$

The Sobol’ indices are defined as follows:(41)$$S_{i_{1},\ldots ,i_{s}}=D_{i_{1},\ldots ,i_{s}}/D$$

By definition, in combination with Equation (39), it is easy to obtain:(42)$$\sum_{i=1}^{n}S_{i}+\sum_{1\leq i<j\leq n}S_{ij}+\ldots +S_{1,2,\ldots ,n}=1$$

Thus, each index $S_{i_{1},\ldots ,i_{s}}$ is a sensitivity measure describing how much of the total variance is due to uncertainty in the set of input parameters $\left\{i_{1},\ldots ,i_{s}\right\}$. The first-order indices $S_{i}$ give the effect of each parameter acting individually on the output, while the second-order indices $S_{ij}$ indicate the coupling effect of variable $x_{i}$ and variable $x_{j}$ on the output, and the higher-order indices describe the effect of a possible mixture of parameters on the output.

The total sensitivity indicators $S_{T_{i}}$ are defined in order to evaluate the total effect of an input variable [43]. They are defined as the sum of all partial sensitivity indices $S_{i_{1},\ldots ,i_{s}}$ containing parameter i:(43)$$S_{T_{i}}=\sum_{\varphi_{i}}S_{i_{1},\ldots ,i_{s}},\;\varphi_{i}=\{\left(i_{1},\ldots ,i_{s}\right):\exists k,1\leq k\leq s,i_{k}=i\}$$

### 3.2. Global Sensitivity Analysis Based on Monte Carlo Simulation

The traditional method of solving the global sensitivity indices is Monte Carlo simulation (MCS). Based on Equations (33) and (39), the following estimates of mean, total and partial variance can be derived using the $N_{MC}$ sample:(44)$${\hat{f}}_{0}=\frac{1}{N_{MC}}\sum_{m=1}^{N_{MC}}f(x_{m})$$
(45)$$\hat{D}=\frac{1}{N_{MC}}\sum_{m=1}^{N_{MC}}f^{2}(x_{m})-{\hat{f}}_{0}^{2}$$
(46)$${\hat{D}}_{i}=\frac{1}{N_{MC}}\sum_{m=1}^{N_{MC}}f(x_{\left(\sim i\right)m}^{\left(1\right)},x_{m}^{\left(1\right)})f(x_{\left(\sim i\right)m}^{\left(2\right)},x_{m}^{\left(1\right)})-{\hat{f}}_{0}^{2}$$
where
(47)$$x_{m}=(x_{1m},x_{2m},\ldots ,x_{nm})\;x_{(\sim i)m}=(x_{1m},x_{2m},\ldots ,x_{\left(i-1\right)m},x_{\left(i+1\right)m},\ldots ,x_{nm})$$

In addition, the superscripts (1) and (2) in Equation (46) indicate that two different samples are generated and mixed. A similar expression allows for a one-time estimation of the total sensitivity indices $S_{T_{i}}$:(48)$${\hat{S}}_{T_{i}}=1-{\hat{D}}_{\sim i}/\hat{D}$$
(49)$${\hat{D}}_{\sim i}=\frac{1}{N_{sim}}\sum_{m=1}^{N_{sim}}f(x_{\left(\sim i\right)m}^{\left(1\right)},x_{m}^{\left(1\right)})f(x_{\left(\sim i\right)m}^{\left(1\right)},x_{m}^{\left(2\right)})-{\hat{f}}_{0}^{2}$$

As described above, global sensitivity analysis does not require any assumptions about the model (e.g., linearity or monotonicity). In practice, analysts usually calculate first-order and total sensitivity indices and sometimes second-order indices. However, the calculation of sensitivity indicators based on the MCS method requires the evaluation of $2^{n}$ integrals, which is not practically feasible unless n is low. In addition, recent work has been devoted to further reducing the computational cost of evaluating Sobol’ indices; see also [46]. However, the computational cost of evaluating all indices through MCS remains an issue.

### 3.3. Global Sensitivity Analysis Based on PCE

Defining a multidimensional indicator $L_{i_{1},\ldots ,i_{s}}$:(50)$$L_{i_{1},\ldots ,i_{s}}=\left\{\alpha \in A:\begin{array}{c} \alpha_{k}>0\;k\in \left(i_{1},\ldots ,i_{s}\right),\forall k=1,\ldots ,n\\ \alpha_{k}=0\;k\notin \left(i_{1},\ldots ,i_{s}\right),\forall k=1,\ldots ,n \end{array}\right\}$$

PCE in Equation (5) can then be rewritten as:(51)$$f_{A}\left(x\right)=\beta_{0}+\sum_{i=1}^{n}\sum_{\alpha \in L_{i}}\beta_{\alpha }\psi_{\alpha }\left(x_{i}\right)+\sum_{1\leq i_{1}<i_{2}\leq n}\sum_{\alpha \in L_{i_{1},i_{2}}}\beta_{\alpha }\psi_{\alpha }\left(x_{i_{1}},x_{i_{2}}\right)+\ldots \;\;\;\;\;\;\;+\sum_{1\leq i_{1}<\cdots <i_{s}\leq n}\sum_{\alpha \in L_{i_{1},\ldots ,i_{s}}}\beta_{\alpha }\psi_{\alpha }\left(x_{i_{1}},{\ldots ,x}_{i_{s}}\right)+\cdots +\sum_{\alpha \in L_{1,\ldots ,n}}\beta_{\alpha }\psi_{\alpha }\left(x\right)$$

Due to the orthogonality of the PC basis, the mean, total variance and partial variance of the response can be easily derived from Equation (51) as:(52)$$\overset{-}{y}=E\left(f\left(x\right)\right)=\beta_{0},\;D^{A}=\sum_{\alpha \in A\backslash \left\{0\right\}}\beta_{\alpha }^{2}E\left[\psi_{\alpha }^{2}\left(x_{i_{1}},{\ldots ,x}_{i_{s}}\right)\right],\;D_{i_{1},\ldots ,i_{s}}^{A}=\sum_{\alpha \in L_{i_{1},\ldots ,i_{s}}}\beta_{\alpha }^{2}\left[\psi_{\alpha }^{2}\left(x_{i_{1}},{\ldots ,x}_{i_{s}}\right)\right]$$

The global sensitivity indices $S_{i_{1},\ldots ,i_{s}}^{A}$ and $S_{i}^{T,A}$ based on PCE obtained from the above equations are expressed as:(53)$$S_{i_{1},\ldots ,i_{s}}^{A}=\frac{D_{i_{1},\ldots ,i_{s}}^{A}}{D^{A}},S_{i}^{T,A}=\sum_{\alpha :\alpha_{i}>0}S_{\alpha }^{A}$$

From the above equations, by modelling the PCE of the response of interest, the global sensitivity indices can be calculated analytically from the coefficients of PCE, which significantly reduces computational costs.

Wu et al. [49] and Cheng et al. [48] derived methods for calculating global sensitivity metrics based on RBF and SVR, respectively. Let us consider the Ishigami function of high nonlinearity and non-monotonicity, which is widely used for benchmarking in global sensitivity analysis:(54)$$Y=\sin X_{1}+a\sin^{2}X_{2}+bX_{3}^{4}\sin X_{1}$$
where a=7, b=0.1 and input variables $X_{i}\left(i=1,2,3\right)$ are uniformly distributed over $\left[-\pi ,\pi \right]$. The sensitivity indices of the model response can be calculated analytically as in [67]. Here, they are approximated by postprocessing PCE, RBF and Kriging of the model response according to [48,49,59].

From Table 4, the sensitivity indices of the parameters can be calculated more accurately using PCE, SVR and RBF for the same number of model evaluations, with the sensitivity indices calculated by PCE being the closest to the theoretical values. Therefore, PCE was subsequently used for the global sensitivity analysis of variables affecting long-term deflection.

## 4. Numerical and Experimental Validations

In this section, we first establish a finite element model of the RC simply supported beam to verify the feasibility of surrogate models with regard to their deflection prediction. Second, the accuracy of different surrogate models in calculating the global sensitivity indices is investigated through a numerical algorithm, and the main variables affecting the maximum deflection of RC simply supported beams are identified through global sensitivity analysis of the geometric and material variables affecting their deflection. Finally, long-term deflection prediction and global sensitivity analysis of RC flexural members are carried out with the collected experimental dataset to further validate the feasibility of surrogate models for application in the field of civil engineering structures.

Figure 1 illustrates the process of prediction and global sensitivity analysis of deflections in RC flexural structures using surrogate models such as PCE, SVR, KRG, and RBF. This process consists of three main steps. In step 1, the dataset is normalised. In step 2, the data are randomly separated into training and testing sets. The training sets are used to train the surrogate model, and the testing sets are used to evaluate models. Step 3 is global sensitivity analysis of the factors affecting the deflection of the RC flexural structure. All experiments were conducted on a desktop computer with a Windows 10 operating system and equipped with an Intel(R) Core(TM) i7-9700 CPU @ 3.00 GHz and 16 GB DDR4 RAM 2666 MHz.

### 4.1. Predictive Accuracy Measures

In this paper, we use the coefficient of determination R^2^, relative average absolute error (RAAE), relative maximum absolute error (RMAE) and root mean square error (RMSE) as evaluation criteria for prediction performance, and these metrics are widely used in the accuracy assessment of surrogate models [68,69,70], the variance accounted factor (VAF), performance index (PI), A_10_−index and uncertainty analysis (U_95_), which are defined as follows:(55)$$R^{2}=1-\frac{\sum_{i=1}^{nt}{\left[f\left(x_{i}\right)-\hat{f}\left(x_{i}\right)\right]}^{2}}{\sum_{i=1}^{nt}{\left[f\left(x_{i}\right)-\overset{-}{f}\right]}^{2}}$$
(56)$$RAAE=\frac{\sum_{i=1}^{nt}\left|f\left(x_{i}\right)-\hat{f}(x_{i})\right|}{nt\times std}$$
(57)$$RMAE=\frac{\max_{1\leq i\leq nt}\left|f\left(x_{i}\right)-\hat{f}(x_{i})\right|}{std}$$
(58)$$RMSE=\sqrt{\frac{\sum_{i=1}^{nt}{\left[f\left(x_{i}\right)-\hat{f}(x_{i})\right]}^{2}}{nt}}$$
(59)$$VAF=(1-\frac{var(f\left(x_{i}\right)-\hat{f}(x_{i}))}{var(f\left(x_{i}\right))})\times 100$$
(60)$$PI=\frac{1}{\left|\overset{-}{f}\right|}\frac{RMSE}{\sqrt{R^{2}}+1}$$
(61)$$A_{10-index}=\frac{m_{10}}{nt}$$
(62)$$U_{95}=\frac{1.96}{nt}\sqrt{\sum_{i=1}^{nt}{\left[f\left(x_{i}\right)-\overset{-}{f}\right]}^{2}+\sum_{i=1}^{nt}{\left[f\left(x_{i}\right)-\hat{f}\left(x_{i}\right)\right]}^{2}}$$
where $f\left(x_{i}\right)$ and $\hat{f}(x_{i})$ are the observed and simulated values, respectively, $\overset{-}{f}$ is the mean of the observed values and nt is the number of samples. Additionally, $m_{10}$ is the number of records with a ratio of measured to predicted value between 0.9 and 1.1. The closer the value of $R^{2}$ and $A_{10-index}$ are to 1, the better the agreement between the actual and predicted values, when the smaller MAAE, RMAE and RMSE and larger VAF show more trustable statistical impressions.

### 4.2. RC Simply Supported Beam

#### 4.2.1. Description of RC Simply Supported Beam Parameters

In this subsection, a RC simply supported beam structure is analysed. The structure and section reinforcement arrangement are shown in Figure 2. The concrete strength grade is C30, and the elastic modulus of concrete and reinforcement are $E_{c}$ and $E_{s}$, respectively. The width and height of the beam section are assumed to be B and H, respectively, the thickness of the concrete protection layer is denoted as a, and the span length of the structure is L. There is a load (denoted as F) applied to the midpoint of the structure. The distribution parameters of all 9 input variables are listed in Table 5. The output *Y* is the midpoint deflection of the RC beam. Additionally, the relationship between two parameters can be calculated using the Parson correlation coefficient (PCC) as:(63)$$\rho_{X,Y}=\frac{cov(X,Y)}{\sigma_{X}\sigma_{Y}}$$
where $\sigma_{X}$ and $\sigma_{Y}$ are the standard deviations of *X* and *Y*, respectively, and cov(X,Y) is the covariance between *X* and *Y.* As shown in Figure 3, high values of positive or negative coefficients affect the accuracy of the model and make it difficult to explain the effect of the input parameters on the target parameters. It can be seen that the correlation between Y and H as well as L and B is very high and that the PCC between the other variables is quite small.

As shown in Figure 4a,b, we established the finite element model (FEM) of a RC simply supported beam in ANSYS 15.0. The FEM analysis was performed by fixing all 9 input variables at their mean values, and the results are shown in Figure 4c. It can be seen that the largest vertical displacements occur in the middle of the simply supported beam. The maximum vertical displacement of the structure is taken as the output of the model and is denoted as Y. The parameters and optimal parameters of the four surrogate models are shown in Table 6.

#### 4.2.2. Results and Discussion

In this numerical example, 100 sample points selected using a Sobol’ quasi-random sequence were used to establish surrogate models. The Sobol’ quasi-random sequence in Ref. [71] with a MALAB implementation called UQLAB is available at http://www.uqlab.com (accessed on 15 May 2023).

Four performance evaluation metrics, namely $R^{2}$, RMAR, RMSE and A10-index, were determined for the above models, and the results for the training and test data are shown in Figure 5a and Figure 5b, respectively. For the training data, Kriging is the optimal model, and for the test data, PCE is the optimal model. Figure 6 further shows the overall ranking of the efficiency of each model in the form of an intuitive stacked graph. Considering both the training and test data, PCE and Kriging are the best, with SVR having the lowest accuracy.

As shown in Table 7, The models were scored from 1 to 4 based on each of the seven indices; then, the scores were summed to assign a total score for each model. As the interpolation-type surrogate models can be able to accurately pass all sample points on the train set, Kriging and RBF have the highest modelling accuracy on the training sets, followed by PCE and SVR. However, the fit-type surrogate models perform better on the testing sets. PCE has the highest prediction accuracy ($R^{2}=0.9907,RAAE=0.0535,RMAE=0.5194,RMSE=0.0115$), followed by Kriging, RBF and SVR. It is noticed that the RMSE prediction accuracies of all 4 surrogate models are below 0.0150.

Figure 7 illustrates the actual-versus-prediction values of the maximum deflection of a RC simply supported beam obtained via PCE, SVR, Kriging and RBF on the same training data and testing data. The closer the data point to the line of best fit, the more accurate the prediction. It can be seen that all 4 surrogate models give particularly good results when predicting the actual deflection values, especially when predicting lower actual deflections; these data are almost always on the line of best fit. The maximum deflection values predicted by the surrogate model for RC beams can reliably support the design process for RC elements. The results of the uncertainty analysis are shown in Table 4. Table 8 shows that for both the training data and test data, all four surrogate models have low U95 values.

Figure 8a–d show the error distributions for the training and testing datasets for the PCE, SVR, Kriging and RBF models. It can be seen that most of the error distributions occur around zero, which leads to high accuracy of the models. All models developed produce more spot distribution around the zero point in the form of a Gaussian bell shape. The Taylor diagram of the surrogate models of a RC simply supported beam is presented in Figure 9. It is seen from these graphs that, despite the excellent performance of all models in high precision, PCE has the best performance in predicting both training and testing data.

#### 4.2.3. Global Sensitivity Analysis of RC Simply Supported Beam

All global sensitivity indices are obtained by post-processing the coefficients of PCE. The results of the MCS method are also listed in Table 7 for comparison and it can be seen that PCE can provide accurate results of all the sensitivity indices with 100 model evaluations.

As shown in Table 9, for the first-order sensitivity indices, it can be seen that the RC simply supported beam section width H has the greatest effect on deflection with $S_{H}=0.4016$, followed by the span length L and load F with $S_{L}=0.3608$ and $S_{F}=0.1040$, respectively. Interestingly, first-order sensitivity indices for the diameter of the tensile reinforcement $d_{2}$ and the thickness of the concrete protective layer a are both 0.0000 with four decimal places retained, which does not mean that they do not have effects on deflection. The total sensitivity indices show that both the diameter of the tensile reinforcement $d_{2}$ and the thickness of the concrete protective layer a have effects on deflection in coupling with other variables. The sum of the first-order sensitivity indices is 0.9433, indicating that the variables acting alone have a dominant effect on deflection, and the sum of the first- and second-order sensitivity indices is 0.9911, close to 1, indicating that there is little higher-order coupling between the variables.

Another interesting result of the global sensitivity analysis is the second-order global sensitivity indices results shown in Figure 10. The x-axis and y-axis are the indices of the variables and the colour denotes the sensitivity indices. The white area indicates no coupling between pairs of variables, and the darker the colour, the greater the value of the second-order sensitivity indices for the pair of variables. It can be seen that in the variable pairs (H,L), (H,F) and (L,F) have very high values, which in fact are correlated. Therefore, the proposed method can also be used as a correlation evaluation tool for the uncertain parameters in the structure.

In summary, this numerical example of an RC simply supported beam demonstrates that all four surrogate models are efficient and accurate in engineering applications. At the same time, this example validates the validity and accuracy of global sensitivity analysis based on PCE in practical engineering applications.

### 4.3. Experiments of Long-Term Deflection of RC Flexural Members

To illustrate the effectiveness of surrogate models on the application of civil engineering problems, the prediction and global sensitivity analysis for long-term deflection tests on RC flexural members is presented in this section.

#### 4.3.1. Data Collection and Pre-Processing

We analysed the data collected from 191 experiments that were summarised and documented by Espion [72] from 29 different research programs. The experimental dataset consists of 181 samples that detail the long-term deflections of RC simply supported beams and slabs with a variety of geometries, load levels and distributions, concrete strengths, reinforcement ratios and environmental conditions. To better evaluate efficiency, the performance of the surrogate models was compared with that of other machine learning models that have been frequently used for solving practical problems related to civil engineering, including back propagation neural networks (BP), decision tree (DT) and linear regression (LR). The hyperparameter settings for BP, DT, LR and the surrogate models were either proposed by previous studies, such as LR by Pham et al. [33], or were the default values for surrogate models.

Table 10 reports the descriptions of 16 input variables and the ultimate long-term defections. The input variables were geometric parameters including (section width (b), total depth (h), area of tensile reinforcement ($A_{s}$), and experimental parameters (distance from ultimate compression fibre to centre of mass of tensile reinforcement (d), tensile reinforcement ratio ($A_{s}/bd$), relative humidity (RH), concrete strength at age t′ ($f_{c}^{\prime }$), span length (l), span-to-depth ratio (l/h), loading age ($t_{i}$), maximum moment at a constant load ($M_{d}$) consisting of the beam’s own weight and a uniform load applied at the same age, maximum moment at an additional continuous load ($M_{q}$) consisting of a concentrated load and a uniform load applied at different ages), factors entering into the elastic deflection equation depending on the static system and load distribution ($K_{d}\;K_{q}$), instantaneous or immediate measured deflection a(i) under $M_{d}+M_{q}$, and age t. The response was total measured deflection a(t) of the concrete flexural structure at age t.

Figure 11 shows the histograms of 17 variables with minima, maxima, mean and standard deviation in the final dataset. Most variables, except element $K_{d}$, $K_{q}$, are well-distributed and suitable for the modelling. As shown in Figure 12, correlations existed except for *RH* and *B* and with *RH* and $K_{q}$ (0.00), with *H* and *d* having the highest correlation (0.99), followed by a(t) with a(i) and l/h at 0.94 and 0.82, respectively. The relationship between the response and the input variables can be expressed as:
(64)$$a\left(t\right)=f(b,h,A,d,\frac{A_{s}}{bd},RH,f_{c}^{\prime },l,\frac{l}{h},t_{i},M_{d},K_{d},M_{q},K_{q},a\left(i\right),t)$$

To better evaluate efficiency, the performance of the surrogate models was compared with that of other machine learning models that have been frequently used for solving practical problems related to civil engineering, including back propagation neural networks (BP), decision tree (DT) and linear regression (LR). The hyperparameter settings for BP, DT, LR and the surrogate models were either proposed by previous studies, such as LR by Pham et al. [33], or were the default values for surrogate models.

The dataset consisting of 197 samples was randomly partitioned into two subsets: the training set with 178 samples (90% of the total dataset) and the test set with the remaining 19 samples (10%). To mitigate the negative effects of attributes with large values, the selected dataset was normalised. As one-time data partitioning is likely to lead to bias, in this study, 10 experiments were conducted using 10 random data partitions, and the comparative surrogate models were run on these data subsets accordingly. Therefore, comparisons of surrogate models were evaluated based on the mean and standard deviation values of the results of the 10 experiments. The parameters and optimal parameters of the four surrogate models are shown in Table 11.

#### 4.3.2. Results and Discussion

Since the experimental data are discrete in nature and contain different levels of noise, a single division of training data and testing data may use some “bad” data as the training data to train the models, resulting in too bad an accuracy of the model on the testing data, so this paper used 10-times attempts to divide training data and testing data to achieve good model training results. The evaluation metrics, namely $R^{2}$, RAAE, RMAE, RMSE, VAF, PI and A_10_-Index, were calculated from the test data to assess the predictive accuracy of surrogate models in predicting the long-term deflection of RC structures. Table 12 lists the values of the metrics calculated by the above models.

Four performance evaluation metrics were identified for the surrogate models, namely $R^{2}$, RMAR, RMSE and A10-index, and the results for the experimental datasets are shown in Figure 13a,b. For the training data, Kriging was the best model, while for the test data, SVR was the best model. Figure 14 further shows the overall ranking of the efficiency of each model in the form of an intuitive stacked graph. Considering both the training and test data, Kriging is the best, with SVR and PCE having the lowest accuracy.

As shown in Table 12, when the training sets were brought into surrogate models for prediction, both Kriging and RBF were able to reconstruct the training sets accurately due to the fact that Kriging and RBF are interpolated surrogate models. When the testing data were brought into the surrogate model for prediction, the prediction accuracy of the fitted surrogate models PCE and SVR was higher than that of Kriging and RBF. The prediction accuracy of SVR is the highest, with evaluation indices of $R_{mean}^{2}=0.9765,R_{std}^{2}=0.0080,{RAAE}_{mean}=0.0965,{RAAE}_{std}=0.0141,{RMAE}_{mean}=0.4463,{RMAE}_{std}=0.1167,{RMSE}_{mean}=0.0555,{RMSE}_{std}=0.0175$.

Figure 15 shows the actual versus predicted values of the maximum deflection of RC simply supported beams obtained via PCE, SVR, Kriging and RBF on the training data and testing data. The closer the data points are to the line of best fit, the more accurate the predicted values are. It can be seen that although RBF has the lowest prediction accuracy, $R^{2}=0.8975$ is able to reach nearly 0.9000. PCE, SVR and Kriging all give good results when predicting the actual deflection values, especially when predicting the lower actual deflection, and the data at these points are almost always on the line of best fit. The maximum deflection values predicted by the surrogate models for RC beams can reliably support the design process for RC members. The results of the uncertainty analysis are shown in Table 13. Table 13 shows that for both the training data and test data, all four surrogate models have low U_95_ values, with the RBF having the lowest (0.0749).

Figure 16a–d show the error distributions for the training and testing datasets for the PCE, SVR, Kriging and RBF models. It can be seen that, consistently with the conclusion for the RC simply supported beam, most of the errors produce a more patchy distribution around the zero point in the form of a Gaussian bell shape. Figure 17 shows Taylor plots of the surrogate models for the experimental data. It can be seen from these plots that RBF and Kriging have the best performance for the training data and SVR has the best performance for the test data.

Figure 18 presents comparisons of the RMSE values that were obtained from the surrogate models, LR model and empirical methods. The surrogate model has much smaller RMSE values than the ACI 318-83 building code and the CEB model code MC78. The RMSE values are also very competitive with the PSO-XGBoost model [73]. Therefore, surrogate models are effective tools for civil engineers or designers in predicting the long-term deflections of RC flexural members.

In order to verify whether the surrogate model is truly better than other models in predicting the long-term deformation of RC beams, a statistical measure of a one-tailed *t*-test statistical measure is performed. RMSE is tested as it is a common error metric for comparing models. The test is carried out on the RMSE values obtained in the test set with an equal number of samples and unequal variance. The calculated results with a confidence level of 95% (α=0.05) are presented in Table 14. For all cases except RBF vs. BP where α is larger than the calculated p value, it indicates that the surrogate model significantly outperformed the other models in terms of RMSE values of the long-term defection prediction of reinforced-concrete beams.

The same conclusion is visually reflected in Figure 19, which shows the box plot of RMSE and $R^{2}$ values yielded by comparative models. Although the LR model exhibits a high $R^{2}$, there is an outlier (+). PCE, SVR and KRG are less variable. The surrogate model demonstrates lower RMSE values and much smaller variability. Therefore, the surrogate model is the best prediction method in this experiment.

#### 4.3.3. Global Sensitivity Analysis of Characteristic Parameters That Affect Long-Term Deflection

Global sensitivity analysis not only identifies the important variables that influence the long-term deformation of RC structures, but also determines the effect of coupling between the variables on the prediction of structural deformation. This subsection addresses the use of global sensitivity analysis based on PCE to provide a comprehensive analysis of the importance of model input variables for predicting the long-term deformation of RC structures. As shown in Figure 20, the rectangular colour blocks on the diagonal line indicate the first-order sensitivity indices. Notably, the variable a(i) has the largest sensitivity index value and the variables H and $K_{q}$ acting alone have no effect on the response. The other variables have smaller first-order sensitivity indices. The results of second-order sensitivity indices show that there is coupling between most of the variables. The sum of all first-order sensitivity indices is $\sum S_{i}=0.9306$ and the sum of the first- and second-order sensitivity indices is $\sum S_{i}+S_{ij}=0.9665$, indicating that there is little higher-order coupling between the variables affecting the long-term deflection of RC flexural members.

Figure 21 shows the total sensitivity indicators for the variables affecting the long-term deflection of RC structures, and it can be clearly seen that all variables are present to influence the response by coupling with other variables, with variable a(i) having the greatest degree of influence on the response by coupling with other variables.

## 5. Conclusions

This paper presents the first prediction of the long-term deformation of RC structures using surrogate models (PCE, SVR, Kriging and RBF). The model accuracy was assessed using the evaluation metrics $R^{2}$, MAAE, RMAE, RMSE, VAF, PI, A_10_-Index and U_95_. and global sensitivity analysis of the parameters affecting the long-term deformation of RC structures was carried out. The feasibility of the proposed method was verified on a numerical example of a RC simply supported beam and a collected experimental dataset. For a RC simply supported beam, PCE has the highest prediction accuracy ($R^{2}=0.9907,RAAE=0.0535,RMAE=0.5194,RMSE=0.0115,VAF=99.0663,PI=0.0249,A_{10}-Index=0.9400$), followed by Kriging, RBF and SVR. It is noticed that the RMSE prediction accuracies of all four surrogate models are below 0.0150. For experiments of long-term deflection of RC flexural members, RBF has the lowest prediction accuracy, with $R^{2}=0.8975$ able to reach nearly 0.9000. PCE, SVR and Kriging all give good results when predicting the actual deflection values, especially when predicting the lower actual deflection, and the data at these points are almost always on the line of best fit. Taylor diagrams show that although all surrogate models have excellent performance in the accurate prediction of RC beam deflection, Kriging and RBF have the best prediction performance for training data and SVR and PCE for testing data. The results of the U_95_ uncertainty analysis show that all four surrogate models have low uncertainty on both the FEM numerical model and the experimental data, with the FEM numerical model having 0.0269 and the experimental data having a maximum value of 0.0753. In addition, the prediction accuracy of the surrogate models are competitive in relation to empirical methods (ACI 318-83 and CEB model code) and machine learning models (PSO-XGBoost, BP, DT and LR).

At the same time, global sensitivity analysis based on PCE is proposed for the first time to determine the most important parameters for predicting the long-term deflection of RC structures. The effects of each factor acting alone or coupled with other factors on the long-term deflection of RC structures are analysed by means of first-order sensitivity indicators and total sensitivity indicators. For a RC simply supported beam, the beam section width H has the greatest effect on deflection with $S_{H}=0.4016$, followed by the span length L and load F with $S_{L}=0.3608$ and $S_{F}=0.1040$, respectively. Additionally, the variable pairs (H,L)*,* (H,F) and (L,F) have very high values, which in fact are correlated. For experiments of long-term deflection of RC flexural members, instantaneous or immediate measured deflection a(i) has the largest sensitivity index value, and all variables are present to influence the response by coupling with other variables, with the variable a(i) having the greatest degree of influence on the response by coupling with other variables.

The results of this paper provide civil engineers and designers with an effective model for predicting the long-term deflection of RC structures and analysing the factors, such as the material and geometric factors, affecting the deflection of concrete beams. The future research directions in structural engineering include the development of user-operated software for the prediction of long-term deflection and global sensitivity analysis based on surrogate models, which are more convenient for engineers to use directly for solving various practical problems. In addition, considering that this paper only focuses on the long-term continuous loading tests of ordinary RC structures, future research will address the use of surrogate models in both high-strength and lightweight concrete materials.

## Figures and Tables

**Figure 1 materials-16-04671-f001:**
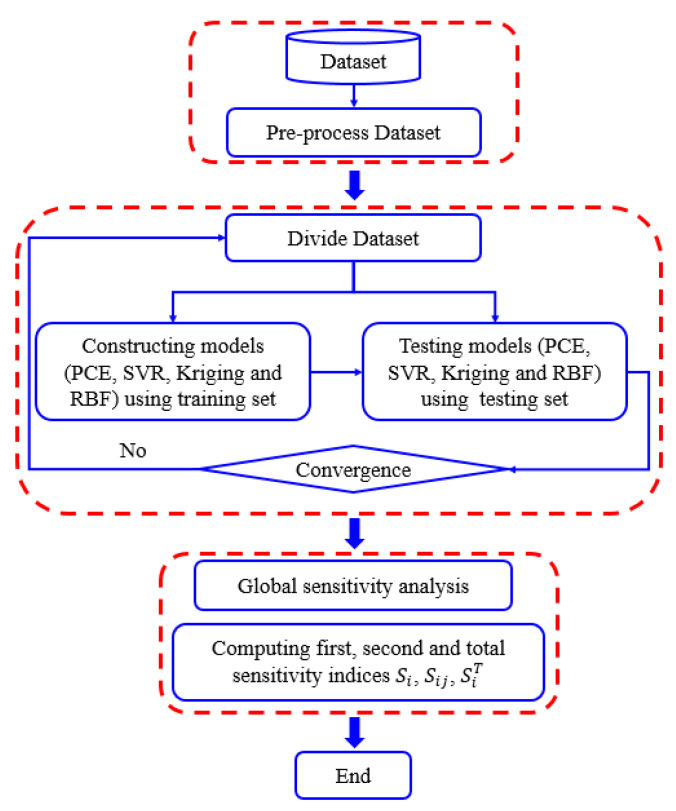
The workflow of prediction and global sensitivity analysis of deflections in RC flexural structures using surrogate models.

**Figure 2 materials-16-04671-f002:**
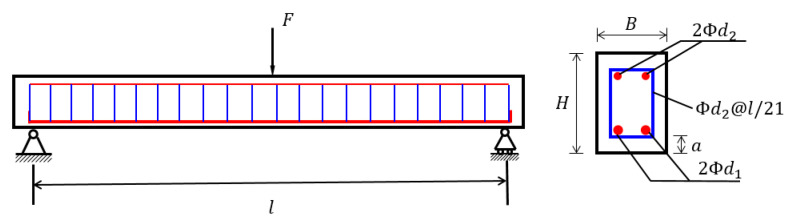
Schematic diagram of a RC simply supported beam.

**Figure 3 materials-16-04671-f003:**
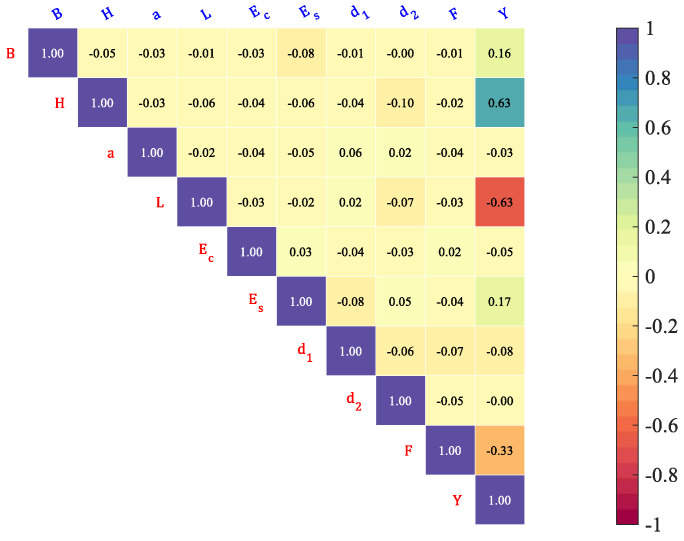
PCC between the variables of a RC simply supported beam.

**Figure 4 materials-16-04671-f004:**
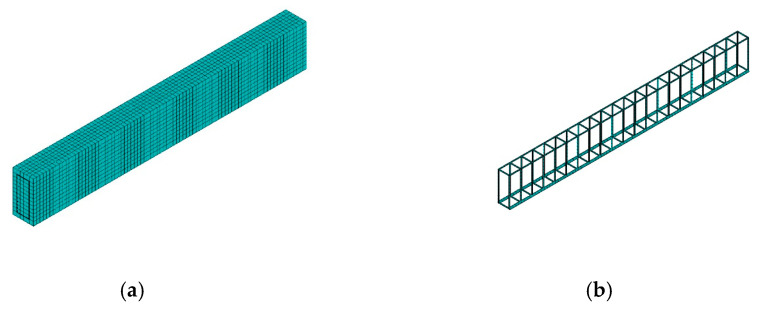
Finite element model of the RC simply supported beam. (**a**) Concrete; (**b**) reinforcement; (**c**) structure deformation diagram.

**Figure 5 materials-16-04671-f005:**
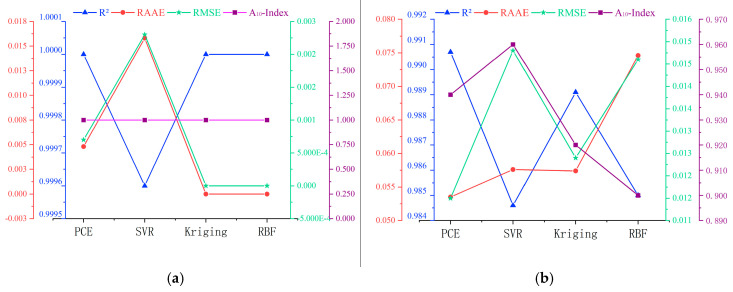
Multi-axis figure of model assessment evaluators for the RC simply supported beam: (**a**) training data, (**b**) testing data.

**Figure 6 materials-16-04671-f006:**
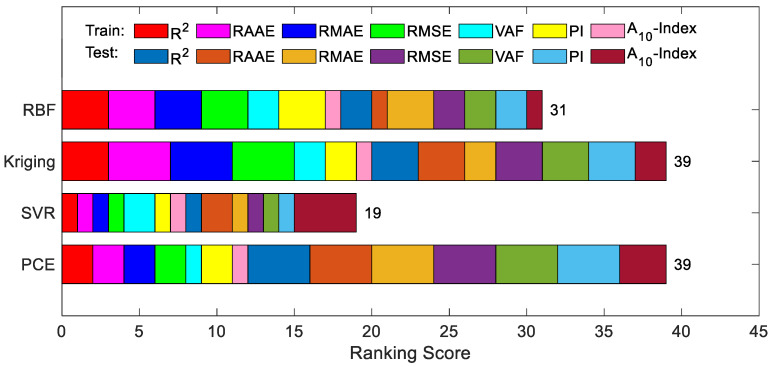
Intuitive presentation of accumulated ranking of developed models on RC beam.

**Figure 7 materials-16-04671-f007:**
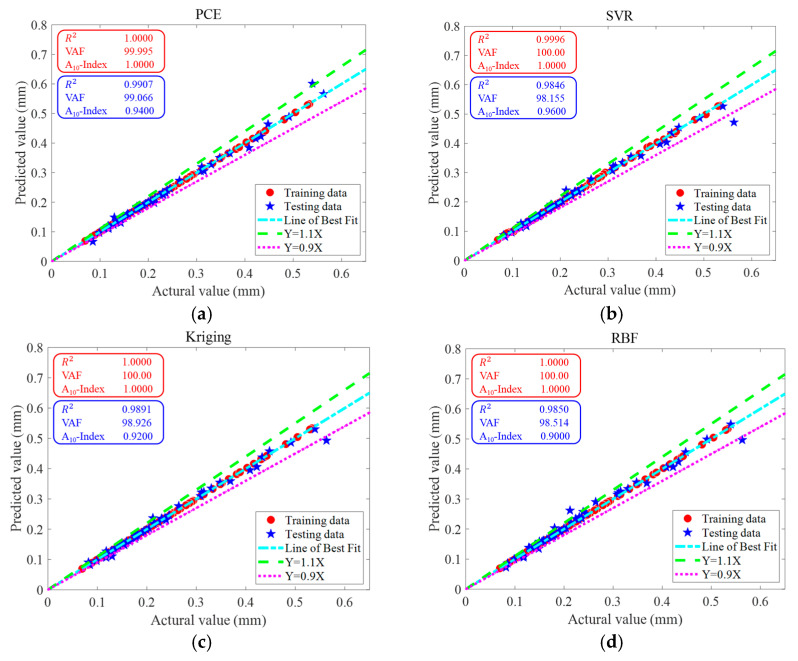
Actual and predicted values of a RC simply supported beam by surrogate models. (**a**) PCE, (**b**) SVR, (**c**) Kriging and (**d**) RBF.

**Figure 8 materials-16-04671-f008:**
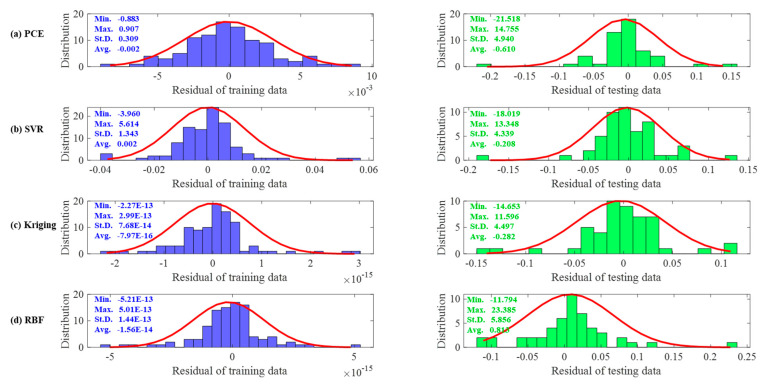
The residual analysis on the training dataset and testing dataset for the RC simply supported beam example.

**Figure 9 materials-16-04671-f009:**
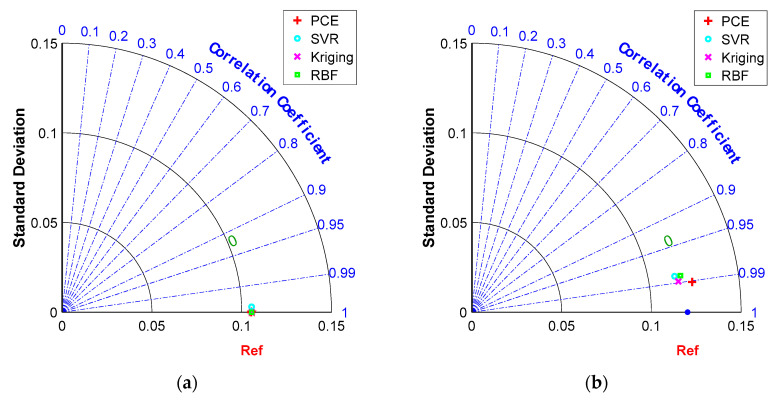
Comparison of model performances in Taylor diagram for the RC simply supported beam example: (**a**) training dataset, (**b**) testing dataset.

**Figure 10 materials-16-04671-f010:**
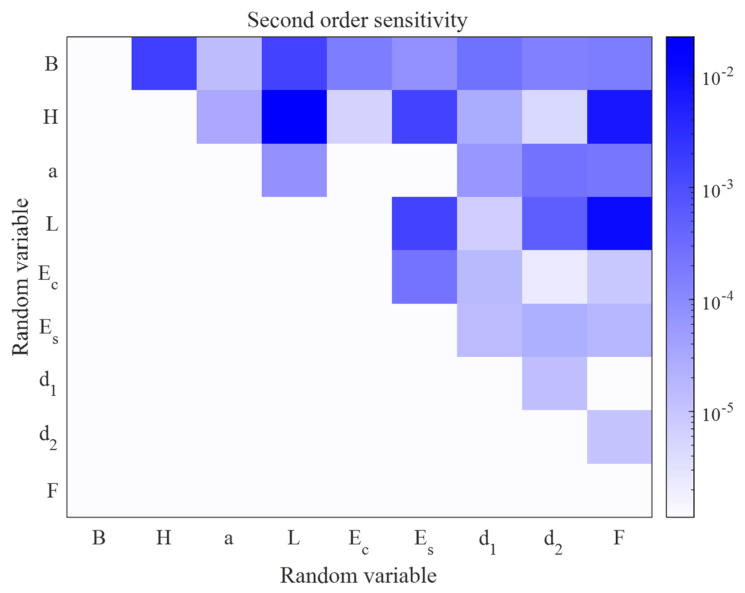
Heatmap visualisation of second-order sensitivity.

**Figure 11 materials-16-04671-f011:**
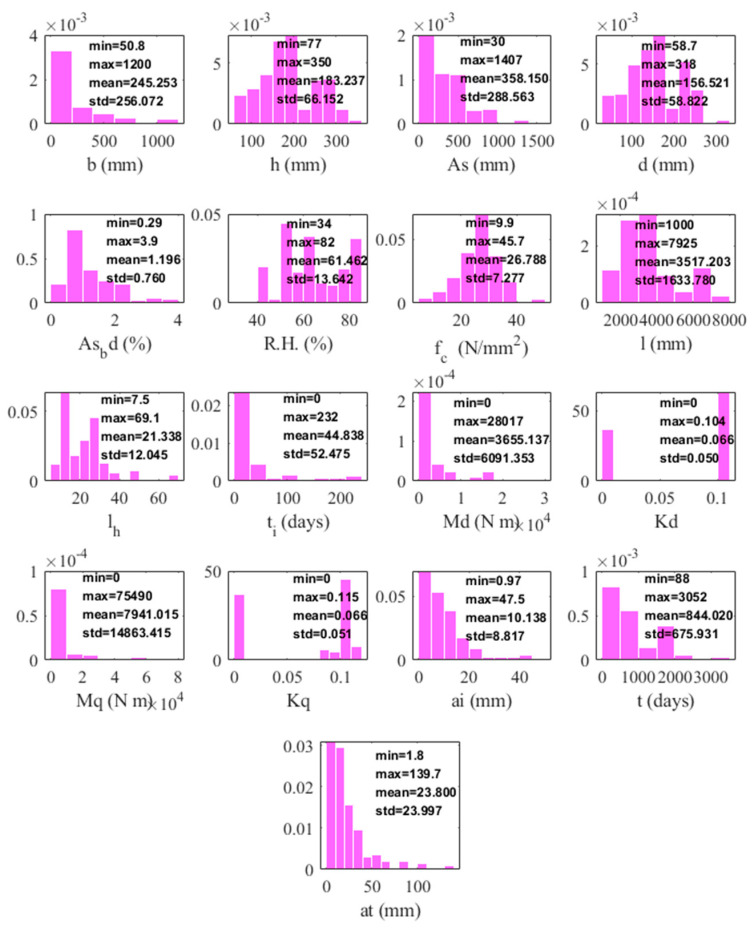
Histograms of the 17 variables in the final dataset (sample count: 197); statistical information such as minimum, maximum, mean, std. are also shown on the histograms.

**Figure 12 materials-16-04671-f012:**
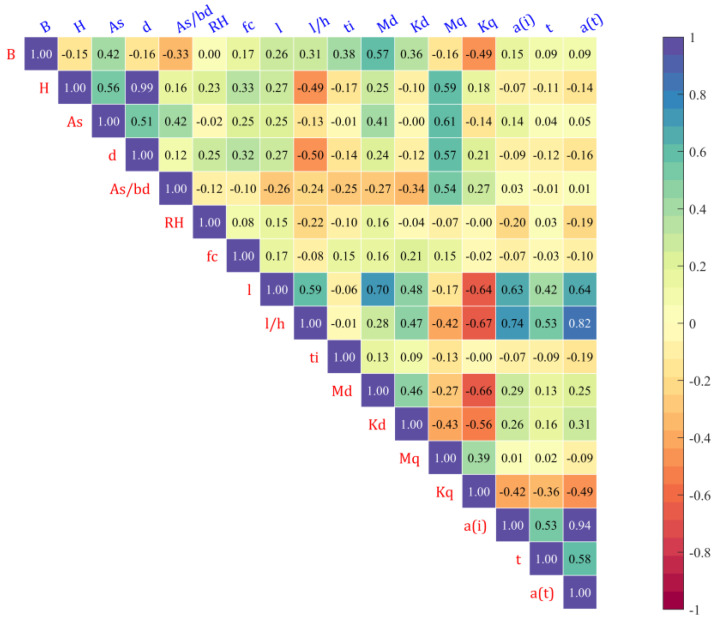
PCC between the variables of the final dataset.

**Figure 13 materials-16-04671-f013:**
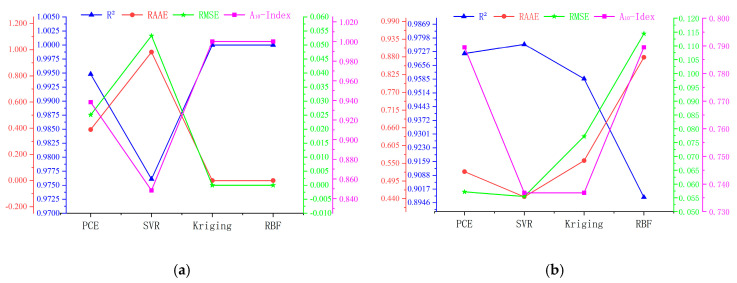
Multi-axis figure of model assessment evaluators: (**a**) training data, (**b**) testing data.

**Figure 14 materials-16-04671-f014:**
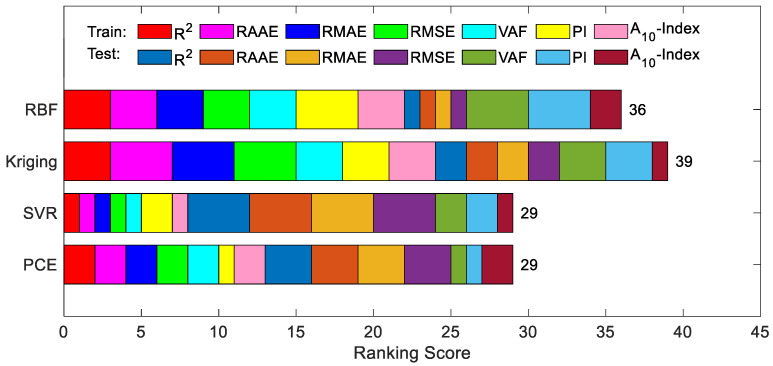
Intuitive presentation of accumulated ranking of developed models on the final dataset.

**Figure 15 materials-16-04671-f015:**
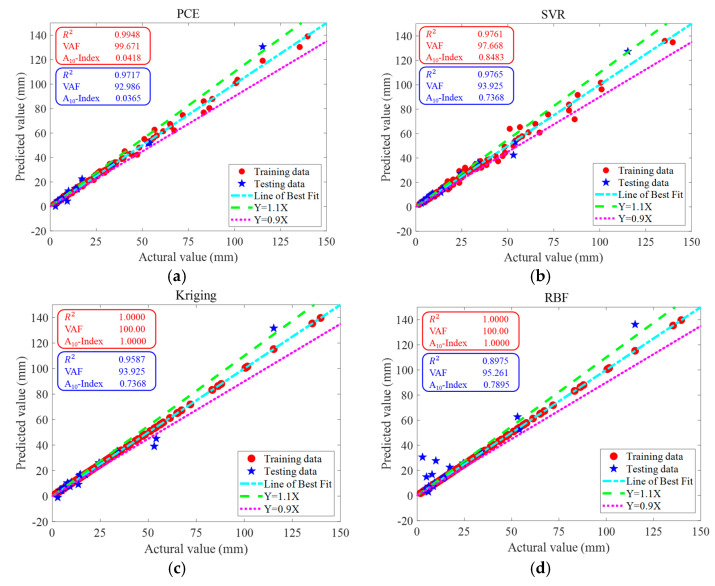
The regression analysis on the training dataset and testing dataset between experimental data (horizontal axis) and surrogate models’ predictions (vertical axis). (**a**) PCE, (**b**) SVR, (**c**) Kriging and (**d**) RBF.

**Figure 16 materials-16-04671-f016:**
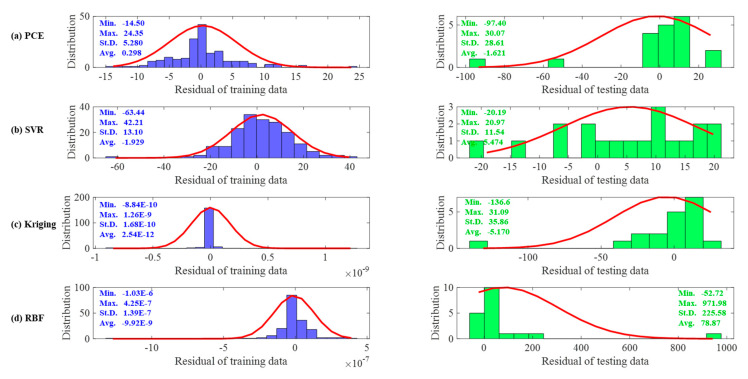
The residual analysis on the experimental data.

**Figure 17 materials-16-04671-f017:**
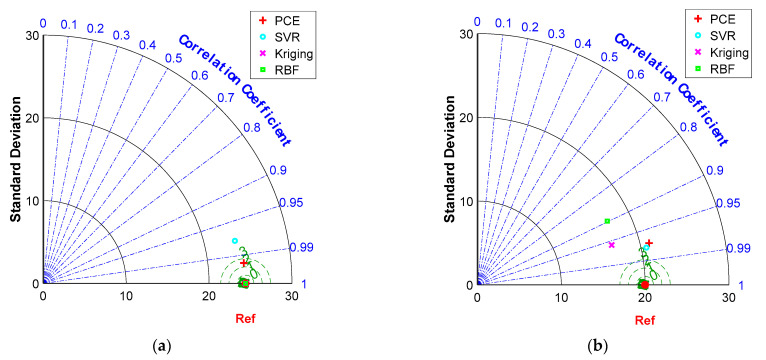
Comparison of model performances in Taylor diagram on the experimental data: (**a**) training dataset, (**b**) testing dataset.

**Figure 18 materials-16-04671-f018:**
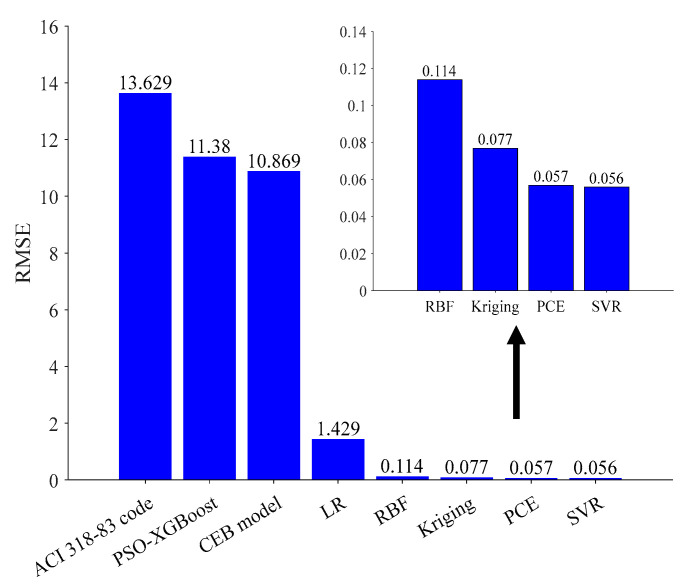
The performance comparison between surrogate model models with LR models and the empirical methods on the experimental data.

**Figure 19 materials-16-04671-f019:**
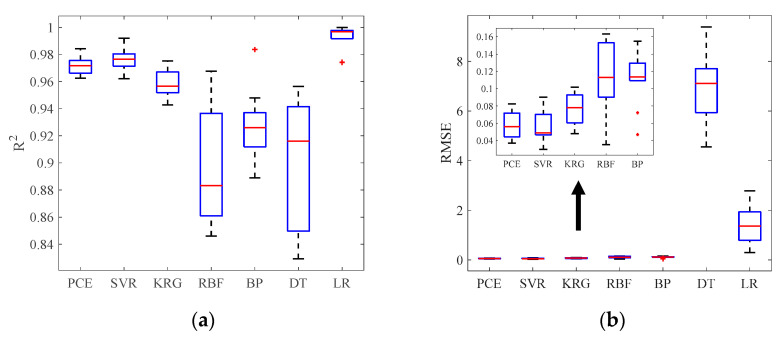
Box plots of prediction models for (**a**) $R^{2}$ and (**b**) RMSE.

**Figure 20 materials-16-04671-f020:**
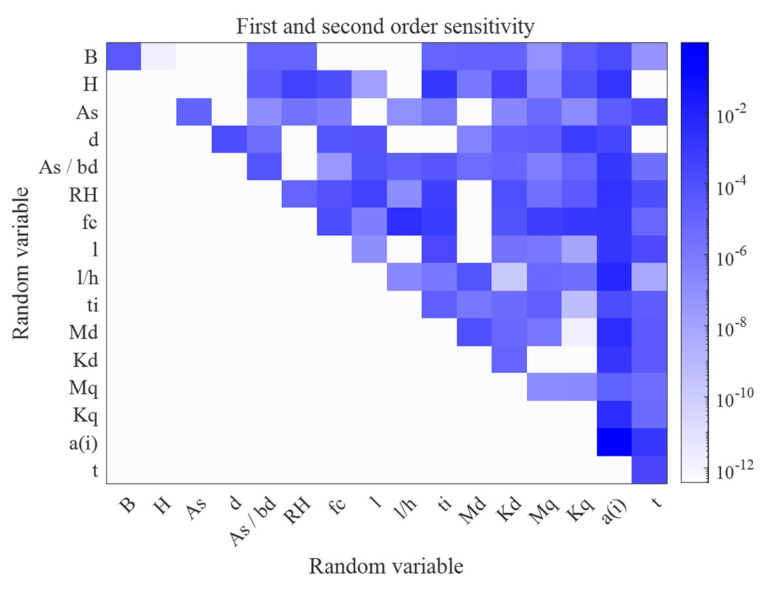
First- and second-order sensitivity for experimental data.

**Figure 21 materials-16-04671-f021:**
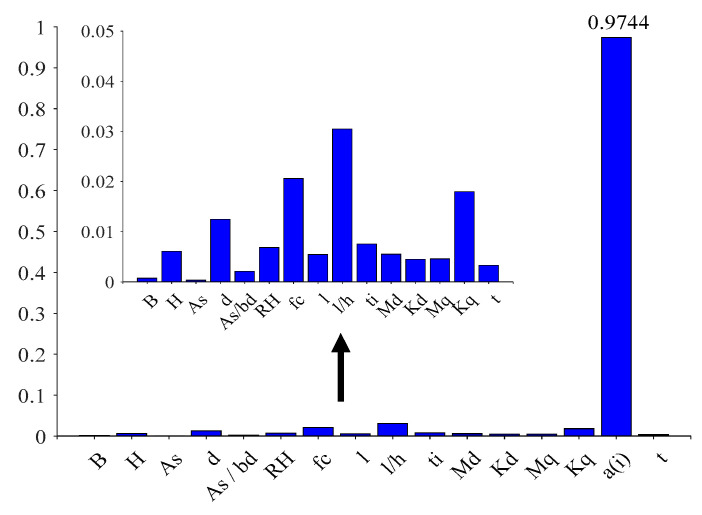
Total sensitivity indices for experimental data.

**Table 1 materials-16-04671-t001:** Common types of orthogonal polynomials and their associated support.

Types	Polynomial	Support
Gaussian	Hermite	$\left(-\infty ,\infty \right)$
Uniform	Legendre	$\left[a,b\right]$
Beta	Jacobi	$\left[a,b\right]$
Gamma	Laguerre	$\left[0,\infty \right)$
Poisson	Charlier	$\left\{0,1,2,\cdots \right\}$
Binomial	Krawtchouk	$\left\{0,1,2,\cdots ,n\right\}$
Negative binomial	Meixner	$\left\{0,1,2,\cdots \right\}$
Hypergeometric	Hahn	$\left\{0,1,2,\cdots ,n\right\}$

**Table 2 materials-16-04671-t002:** Some commonly used kernel functions for SVR.

Kernel	Description
Linear kernel	$\phi \left(x_{i},x_{j}\right)=x_{i}^{T}x_{j}$
Polynomial kernel	$\phi \left(x_{i},x_{j}\right)={\left(x_{i}^{T}x_{j}+t\right)}^{d},t\geq 0$
Radial basis kernel	$\phi \left(x_{i},x_{j}\right)=e^{-{\left\|x_{i}-x_{j}\right\|}^{2}/2\sigma^{2}}$

**Table 3 materials-16-04671-t003:** Some commonly used basis functions for RBF.

Types	Expressions
Multiquadric basis (MQ)	$\phi \left(r\right)=\sqrt{r^{2}+c^{2}}$
Inverse multiquadric basis (IMQ)	$\phi \left(r\right)=1/\sqrt{r^{2}+c^{2}}$
Gaussian basis (G)	$\phi \left(r\right)=e^{-r^{2}/c^{2}}$
Thin plate spline (TPS)	$\phi \left(r\right)=r^{2}\ln\left(r+c\right)$

**Table 4 materials-16-04671-t004:** Sensitivity analysis results for Ishigami function.

Sensitivity Indices	Analytical Results	PCE	Error (%)	SVR	Error (%)	RBF	Error (%)	MCS	Error (%)
$S_{1}$	0.31	0.31	0.06	0.31	0.2	0.32	2.2	0.35	0.25
$S_{2}$	0.44	0.44	0.02	0.45	0.9	0.38	14.4	0.44	0.34
$S_{3}$	0	0	0	0	-	0.00	-	0.00	-
$S_{13}$	0.24	0.24	0.04	0.24	1.4	0.22	8.3	0.24	0.04
$S_{1}^{T}$	0.57	0.56	1.71	0.55	3.5	0.58	2.2	0.56	0.30
$S_{2}^{T}$	0.44	0.44	0.02	0.45	0.9	0.50	12.5	0.44	0.06
$S_{3}^{T}$	0.24	0.24	0.04	0.24	1.4	0.28	15.6	0.24	0.04
Model evaluations		65		65		65		$1.0\times 10^{6}$	

**Table 5 materials-16-04671-t005:** Statistical details of input variables and values for a RC simply supported beam.

No.	Variable	Description	Distribution	Mean	C.O.V.
1	B	Section width (mm)	Normal	150	0.1
2	H	Total depth (mm)	Normal	300	0.1
3	a	Thickness of concrete cover	Normal	30	0.1
4	L	Span length (mm)	Normal	2000	0.1
5	$E_{c}$	Elasticity modulus of concrete (MPa)	Normal	24,000	0.1
6	$E_{s}$	Elastic modulus of the reinforcement (MPa)	Normal	200,000	0.1
7	$d_{1}$	Radius of the tensile bar (mm)	Normal	16	0.1
8	$d_{2}$	Radius of the stirrup and compression bar (mm)	Normal	8	0.1
9	F	Load (KN)	Normal	80	0.15

**Table 6 materials-16-04671-t006:** The parameters of the surrogate models and the optimal parameters for the RC simply supported beam.

Surrogate Models	Expression	Optimal Value
PCE	Polynomial of order p	3
SVR	Tuning parameter σ	6.2766
Kriging	Initial correlation coefficient θ	1.6
RBF	Type of basis function	TPS

**Table 7 materials-16-04671-t007:** The results of surrogate model accuracies for the RC simply supported beam example.

Criteria	Training				Testing			
	PCE	SVR	Kriging	RBF	PCE	SVR	Kriging	RBF
$R^{2}$	1	0.9996	1	1	0.9907	0.9846	0.9891	0.9850
Score	2	1	3	3	4	1	3	2
RAAE	0.0048	0.0158	8.65 × 10^−16^	1.59 × 10^−15^	0.0535	0.0576	0.0574	0.0746
Score	2	1	4	3	4	2	3	1
RMAE	0.0213	0.0694	3.44 × 10^−15^	7.40 × 10^−15^	0.5194	0.7656	0.5890	0.5580
Score	2	1	4	3	4	1	2	3
RMSE	0.0007	0.0023	1.15 × 10^−16^	2.23 × 10^−16^	0.0115	0.0148	0.0124	0.0146
Score	2	1	4	3	4	1	3	2
VAF	99.9952	100	100	100	99.0663	98.1545	98.9257	98.5140
Score	1	2	2	2	4	1	3	2
PI	0.0016	0.0049	2.56 × 10^−16^	4.94 × 10^−16^	0.0249	0.0353	0.0269	0.0316
Score	2	1	2	3	4	1	3	2
A10-Index	1	1	1	1	0.9400	0.9600	0.9200	0.9000
Score	1	1	1	1	3	4	2	1
Total score	12	8	20	18	27	11	19	13

**Table 8 materials-16-04671-t008:** Comparisons of the performance results for U_95_ uncertainty for the RC simply supported beam example.

Model	PCE	SVR	Kriging	RBF
Training data	0.0206	0.0206	0.0206	0.0206
Testing data	0.0331	0.0333	0.0332	0.0332
Average	0.0269	0.0269	0.0269	0.0269

**Table 9 materials-16-04671-t009:** Results of global sensitivity analysis of input variables affecting the deflection of the RC simply supported beams.

First-Order Sensitivity Indices			Total Sensitivity Indices		
	PCE	MCS		PCE	MCS
$S_{B}$	0.0383	0.0383	$S_{B}^{T}$	0.0425	0.0425
$S_{H}$	0.4016	0.4015	$S_{H}^{T}$	0.4401	0.4405
$S_{a}$	0.0000	0.0000	$S_{a}^{T}$	0.0009	0.0009
$S_{L}$	0.3608	0.3605	$S_{L}^{T}$	0.4012	0.4013
$S_{E_{c}}$	0.0001	0.0001	$S_{E_{c}}^{T}$	0.0023	0.0023
$S_{E_{s}}$	0.0385	0.0385	$S_{E_{s}}^{T}$	0.0461	0.0461
$S_{d_{1}}$	0.0001	0.0001	$S_{d_{1}}^{T}$	0.0006	0.0006
$S_{d_{2}}$	0.0000	0.0000	$S_{d_{2}}^{T}$	0.0018	0.0018
$S_{F}$	0.1040	0.1037	$S_{F}^{T}$	0.1302	0.1301
Model evaluations	100	$10^{6}\times 10$ (average)	-	100	$10^{6}\times 10$ (average)
$\sum S_{i}$	0.9433		$\sum S_{i}+S_{ij}$	0.9911	

**Table 10 materials-16-04671-t010:** Descriptions of dataset for RC flexural members.

No.	Variables	Attribute
1	b	Section width (upper or compressed fibre) (mm)
2	h	Total depth (mm)
3	$A_{s}$	Area of tensile reinforcement (${mm}^{2}$)
4	d	Distance from the extreme compression fibre to centroid of tension reinforcement (mm)
5	$A_{s}/bd$	Tensile reinforcement ratio ρ
6	RH	Relative humidity (given or assumed) (%)
7	$f_{c}^{\prime }$	Concrete strength at age t′ (${N/mm}^{2}$)
8	l	Span length (mm)
9	l/h	Span/depth ratio
10	$t_{i}$	Age at loading (days)
11	$M_{d}$	Maximum bending moment due to dead load constituted by the beam’s own weight and by the uniform sustained loading if applied at the same age (N·m)
12	$K_{d}$	Factors entering elastic deflection formulae by the beam’s own weight and by the uniform sustained loading if applied at the same age
13	$M_{q}$	Maximum moment due to additional sustained loading constituted by concentrated loading or by uniform loading if applied at a different age than the dead load (N·m)
14	$K_{q}$	Factors entering elastic deflection formulae due to additional sustained loading constituted by concentrated loading or by uniform loading if applied at a different age than the dead load
15	a(i)	Instantaneous or immediate measured deflection under $M_{d}+M_{q}$ (mm)
16	t	Age (days)
17	a(t)	Total measured deflection at age t under $M_{d}+M_{q}$ (mm)

**Table 11 materials-16-04671-t011:** The parameters of the surrogate models and the optimal parameters.

Surrogate Models	Expression	Optimal Value
PCE	Polynomial of order p	5
SVR	Tuning parameter σ	3.2036
Kriging	Initial correlation coefficient θ	97.37
RBF	Type of basis function	MQ

**Table 12 materials-16-04671-t012:** Results of surrogate model prediction performance.

Criteria	Training				Testing			
	PCE	SVR	Kriging	RBF	PCE	SVR	Kriging	RBF
$R^{2}$	0.9948	0.9761	1	1	0.9717	0.9765	0.9587	0.8975
Score	2	1	3	3	3	4	2	1
RAAE	0.0367	0.0896	3.10 × 10^−13^	4.98 × 10^−10^	0.1064	0.0965	0.1401	0.2072
Score	2	1	4	3	3	4	2	1
RMAE	0.3898	0.9814	1.11 × 10^−11^	2.28 × 10^−9^	0.5242	0.4463	0.5583	0.8783
Score	2	1	4	3	3	4	2	1
RMSE	0.0251	0.0533	4.59 × 10^−13^	2.20 × 10^−10^	0.0572	0.0555	0.0772	0.1144
Score	2	1	4	3	3	4	2	1
VAF	99.6713	97.6675	100.0000	100.0000	92.9864	93.9251	94.3046	95.2610
Score	2	1	3	3	1	2	3	4
PI	0.0418	0.0391	0.0378	0.0364	0.0365	0.0340	0.0330	0.0317
Score	1	2	3	4	1	2	3	4
A_10_-Index	0.9382	0.8483	1.0000	1.0000	0.7895	0.7368	0.7368	0.7895
Score	2	1	3	3	2	1	1	2
Total score	13	8	24	22	16	21	15	14

**Table 13 materials-16-04671-t013:** Comparisons of the performance results for U_95_ uncertainty.

Model	PCE	SVR	Kriging	RBF
Training data	0.0525	0.0531	0.0525	0.0525
Testing data	0.0981	0.0976	0.0975	0.0973
Average	0.0753	0.0753	0.0750	0.0749

**Table 14 materials-16-04671-t014:** Calculated *p* values of the one-tailed *t* test measure.

Pairwise Comparisonof RMSE Results	p Value	Test Result
PCE vs. BP	0.0002	Reject $H_{0}$
PCE vs. DT	0.0000	Reject $H_{0}$
PCE vs. LR	0.0001	Reject $H_{0}$
SVR vs. BP	0.0001	Reject $H_{0}$
SVR vs. DT	0.0000	Reject $H_{0}$
SVR vs. LR	0.0001	Reject $H_{0}$
RBF vs. BP	0.5335	Accept $H_{0}$
RBF vs. DT	0.0000	Reject $H_{0}$
RBF vs. LR	0.0002	Reject $H_{0}$

Notes: $H_{0}$: null hypothesis, ${RMSE}_{surrogate\;models}-{RMSE}_{other\;models}\leq 0$; $H_{1}$*: alternative hypothesis, *
${RMSE}_{surrogate\;models}-{RMSE}_{other\;models}>0$.

## Data Availability

Data will be made available on request.

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
