# Peer review of "Prediction and Global Sensitivity Analysis of Long-Term Deflections in Reinforced Concrete Flexural Structures Using Surrogate Models"

_materials, 2023, doi:10.3390/ma16134671_

Round 1
Reviewer 1 Report
Please find the attachment.

Moderate editing of English language is needed.
Reviewer 2 Report
This is an interesting work; however, before proceeding to the next step, the authors should address the following comments.
1. The language of the manuscript has to be improved.
2. Provide a more in-depth discussion of related previous works.
3. Authors should also provide more meaningful discussions regarding the repeatability and reproducibility of the conducted tests/analysis.
4. In the “Conclusion” section, I recommend presenting more quantitative data as the main results of the research study.
The language of the manuscript has to be improved.
Reviewer 3 Report
The article is well written and shows interesting results according to topic.
Just a few observation were done, which are in the attached file.
Review: materials-2425216
Title: The prediction and global sensitivity analysis of long-term deflections in reinforced concrete flexural structures using surrogate models.
Authors: Wenjiao Dan, et. al.
Abstract
Abstract is well redacted. However, I would like authors could define at the beginning that is a Reinforced Concrete (RC).
Introduction
This section is well described and have all element necessary to understand the topic.
Theoretical bases of surrogate model
This section is well explained according all the models proven and examined.
Numerical and experimental validations
In lines 417 to 422, at what values the authors are referring exactly?
What are the observed and simulated values?
Results and Discussion
Line 632, there is a point instead a comma between "respectively" and "for". Or the word "for" must to beginning with capital letter.
Line 647. Agian There is a point and the next word starts with lowercase.
Final question
Just as a doubt, considering a reinforced concrete with an internal structure that gives it that character, is this type of common reinforcement? And if another type of material is added to the concrete, for example waste from demolition and construction, are these models still valid, considering fine and coarse aggregates?
Thanks
Author Response
Response to the Reviewer’ Comments
The authors appreciate very much the comments and suggestions from the Special Issue Editor and Reviewers. The suggestions are very helpful to improve the paper and incorporated in the revised manuscript (Round 1: changes in yellow highlights, Round 2 : changes in green highlights). Our response and revisions are summarized as follows.
Comments to the Author
The article is well written and shows interesting results according to topic.
Just a few observation were done, which are in the attached file.
Review: materials-2425216
Title: The prediction and global sensitivity analysis of long-term deflections in reinforced concrete flexural structures using surrogate models.
Authors: Wenjiao Dan, et. al.
Authors’ response:
We thank the reviewer for this affirmation.
Abstract
- Abstract is well redacted. However, I would like authors could define at the beginning that is a Reinforced Concrete (RC).
Authors’ response:
According to the reviewers' comments, we have added the definition of Reinforced Concrete (RC) to Abstract, as follows:
Line 12-13:
Reinforced concrete (RC) is the result of a combination of steel reinforcing rods (which have high tensile) and concrete (which have high compressive strength).
Introduction
- This section is well described and have all element necessary to understand the topic.
Authors’ response:
We thank the reviewer for the favorable comment.
Theoretical bases of surrogate model
- This section is well explained according all the models proven and examined.
Authors’ response:
We thank the Reviewer for the positive comment.
Numerical and experimental validations
- In lines 417 to 422, at what values the authors are referring exactly?
Authors’ response:
Typically, the observed values are the results of finite element calculations or experimental observations (also named the actual values) and the simulated values are the values obtained using surrogate models (also named the prediction values).
The statistical metrics (R2, RAAE, RMAE, RMSE, VAF, PI, A10-index and U95) in this manuscript are used to thoroughly assess the accuracy of surrogate models. For example, the closer the observed values are to the simulated values, the closer the R2 is to 1, which means that the accuracy of surrogate models is higher.
- What are the observed and simulated values?
Authors’ response:
In this manuscript, the observed values refer to the mid-point deflection of the finite element model of the RC beam in the first example and to the long-term deflections of RC flexural structures obtained experimentally in the second example. Simulated values refer to values obtained by surrogate models (i.e.: PCE, SVR, Kriging and RBF) in both examples.
Results and Discussion
- Line 632, there is a point instead a comma between "respectively" and "for". Or the word "for" must to beginning with capital letter.
Authors’ response:
Thanks for the reviewer’s careful insights. We apologize for this negligence and have corrected the grammatical error. Please see line 631 in the revised manuscript by green highlighting. As follows:
Line 630-633:
Four performance evaluation metrics were identified for the surrogate models, namely , RMAR, RMSE and A10-index, and the results for the experimental datasets are shown in Figures 13(a) and 13(b) respectively. For the training data, Kriging was the best model, while for the test data, SVR was the best model.
- Line 647. Agian There is a point and the next word starts with lowercase.
Authors’ response:
Thanks for the reviewer’s careful insights. We apologize for this negligence and have corrected the grammatical errors. Please see line 647 in the revised manuscript by green highlighting. As follows:
Line 645-650:
When the testing data was brought into surrogate model for prediction, the prediction accuracy of the fitted surrogate models PCE and SVR was higher than that of Kriging and RBF. The prediction accuracy of SVR is the highest, with evaluation indices .
Final question
- Just as a doubt, considering a reinforced concrete with an internal structure that gives it that character, is this type of common reinforcement? And if another type of material is added to the concrete, for example waste from demolition and construction, are these models still valid, considering fine and coarse aggregates?
Thanks
Authors’ response:
There are many common types of reinforcement in actual construction projects, and in this manuscript we use a common form of reinforcement as an example in our numerical example to verify the accuracy and feasibility of the proposed surrogate models and global sensitivity analysis method. The experimental data collected containing the area of tensile reinforcement and the rate of tensile reinforcement further validates the applicability of the proposed method.Surrogate models, as a data-driven modelling approach, are similar to machine learning models in that the problem investigated is viewed as a black-box function, modelled by the relevant input variables and outputs of interest, without concern for the internal details of the problem. So we believe that if another material is added to the concrete, for example waste from demolition and construction, the surrogate models are still valid, considering fine and coarse aggregates. It is possible that, as the complexity of the analytical problem increases, the accuracy of the current surrogate models decreases. Therefore, surrogate models need to be improved so that their accuracy meets the needs of practical engineering applications. For example, dimensionality reduction techniques (i.e. PCA, POD and subspace methods, etc.) can be combined with surrogate models to deal with high-dimensional problems involving a large number of variables and data. This is precisely the problem that our subsequent research will address.
The authors thank the Editor and Reviewers again for the compliment and many useful suggestions to make the manuscript better and clearer.

Round 2
Reviewer 1 Report
Considerable efforts of the authors is appreciated. The revised paper has addressed all my previous comments, and I suggest to ACCEPT the paper as it is now.
Minor editing of English language is required.
Author Response
We really appreciate this positive comment. During the revision period, the manuscript has undergone English language editing by MDPI.